# Learning Guarantees for Non-convex Pairwise SGD with Heavy Tails

## Abstract

In recent years, there have been a growing number of works studying the generalization properties of pairwise stochastic gradient descent (SGD) from the perspective of algorithmic stability. However, few of them devote to simultaneously studying the generalization and optimization for the non-convex setting, especially the ones with heavy-tailed sub-Weibull gradient noise. This paper establishes the stability-based learning guarantees for non-convex, sub-Weibull pairwise SGD by investigating its generalization and optimization jointly. Firstly, we bound the generalization error of pairwise SGD in the general non-convex setting, after bridging the quantitative relationships between $\ell_1$ on-average model stability and generalization error. Secondly, a refined generalization bound is established for non-convex pairwise SGD by introducing the sub-Weibull gradient noise to remove the bounded gradient assumption. Finally, the sharper error bounds for generalization and optimization are provided under the gradient dominance condition. In addition, we extend our analysis to the corresponding pairwise minibatch SGD and derive the first stability-based near-optimal generalization and optimization bounds which are consistent with many empirical observations. These theoretical results fill the learning theory gap for non-convex pairwise SGD with the sub-Weibull tails.

## 1 Introduction

Pairwise learning has attracted much attention in machine learning literature, where its prediction performance is measured by pairwise loss function. Typical paradigms of pairwise learning include metric learning (Xing et al., 2002; Jin et al., 2009; Weinberger & Saul, 2009; Cao et al., 2016), ranking (Clémençon et al., 2008; Agarwal & Niyogi, 2009; Rejchel, 2012), AUC maximization (Cortes & Mohri, 2003; Gao et al., 2013; Ying et al., 2016; Liu et al., 2018), gradient learning (Mukherjee & Zhou, 2006), and learning under the minimum error entropy criterion (Príncipe, 2010; Hu et al., 2015). Despite enjoying the benefits of particular contrastive motivations, pairwise learning often suffers from a heavy computational burden as its optimization objective involves $\mathcal{O}\left(n^2\right)$ terms for the problems with $n$ training samples.

It is well known that stochastic gradient descent (SGD) is ubiquitous and popular for deploying learning systems due to its low time complexity (Lei et al., 2021b) and high adaptability to big data (Bottou & Bousquet, 2007; Lei & Ying, 2020). As a natural extension of SGD, minibatch SGD iteratively updates the model parameter based on several selected samples rather than a single sample, which can further reduce the variance and accelerate algorithmic convergence (Cotter et al., 2011; Dekel et al., 2012; Yin et al., 2018). Therefore, it is natural to employ SGD and minibatch SGD to formulate the computing procedure of pairwise learning. Along with the wide applications of SGD and minibatch SGD in pairwise learning, there are some theoretical progresses focusing on their generalization guarantees recently (Lei et al., 2021b; 2020; Shen et al., 2019; Yang et al., 2021). However, most of the existing theoretical results are limited to convex losses, which can not cover typical pairwise learning algorithms with non-convex losses, e.g., neural networks-based pairwise learning (Huang et al., 2017; Köppel et al., 2019; Li et al., 2022).

Moreover, most of the previous theoretical works of pairwise SGD and its variants require the bounded variance condition (Zhou et al., 2022) and the sub-Gaussian tail assumption limiting the tail performance of the gradient noise (Simsekli et al., 2019a;b). However, these assumptions may be too

idealistic in practice. Indeed, SGD may involve the extremely large variance but the bounded $p$-th moment for some $p \in (1, 2]$ (Cutkosky & Mehta, 2021), and it often shows the heavier-tailed gradient noise in many learning problems (Gorbunov et al., 2020; Madden et al., 2020; Gürbüzbalaban et al., 2021; Lei & Tang, 2021; Li & Liu, 2022). For example, Simsekli et al. (2019b) studied the statistical characteristics of gradient noise of SGD and stated that the gradient noise expresses a heavy-tailed behavior under an isotropic model (also see Zhou et al. (2020); Zhang et al. (2020b)). The heavy-tailed gradient noise may degrade the generalization performance of SGD methods significantly (Nguyen et al., 2019; Hodgkinson & Mahoney, 2021). However, Raj et al. (2023a;b) found that heavy tails of gradient noise can help with generalization in pointwise SGD. Therefore, it is crucial to investigate the theoretical guarantees of non-convex pairwise SGD with heavy tails. As far as we know, this issue has been rarely touched for pairwise SGD.

Bottou & Bousquet (2007) demonstrated that the model performance depends on the joint influence of generalization error and optimization error. The generalization error is used to evaluate the performance of a trained model to some unseen inputs (Vapnik, 1998) and the optimization error concerns the gap between the actual empirical risk and the optimal empirical risk (Li & Liu, 2022). Hence, it is necessary to investigate both generalization error and optimization error for a better understanding of the learning guarantees of SGD. Following this line, some error analysis of SGD can be found in Lei et al. (2020; 2021a). Compared with uniform convergence analysis for SGD methods (Lei et al., 2021b; Lei & Tang, 2021; Li & Liu, 2022; Foster et al., 2018), algorithmic stability analysis often enjoys promising properties on adaptability (Agarwal & Niyogi, 2009; Hardt et al., 2016; Xing et al., 2021) and flexibility (Lei et al., 2021b; Lei & Ying, 2020). In particular, the stability-based theory analysis is suitable for wide application scenarios and independent of the capacity of hypothesis function space (Zhou et al., 2022; Hardt et al., 2016; Bousquet & Elisseeff, 2002; Shalev-Shwartz et al., 2010).

At present, stability and generalization have been well characterized for the non-convex pointwise SGD (Zhou et al., 2022; Hardt et al., 2016). This paper develops the previous analysis techniques (Lei et al., 2021b; Lei & Ying, 2020; Madden et al., 2020; Li & Liu, 2022) to the sub-Weibull pairwise cases by considering the generalization and optimization errors simultaneously. The main contributions of this paper are summarized as follows:

- *Generalization bounds of non-convex pairwise SGD*. After bridging the $\ell_1$ on-average model stability and generalization error, we establish the stability-based generalization bounds for non-convex pairwise SGD. Even under the general non-convex setting, our derived result is comparable with previous works for convex pairwise SGD (Lei et al., 2021b; 2020; Yang et al., 2021). Moreover, the refined bounds are stated by introducing the sub-Weibull gradient noise assumption, where the standard requirement of the bounded gradient is removed.

- *Learning guarantees of non-convex pairwise SGD with gradient dominance condition*. Sharper bounds for generalization error and excess risk are provided for the non-convex pairwise SGD under an additional gradient dominance condition. The current analysis extends the previous analysis for pointwise SGD with sub-Weibull tails (Li & Liu, 2022) to the complicated pairwise setting, and shows the competitive learning rates. Finally, we develop our analysis to the corresponding minibatch case and give the first-ever-known stability-based learning guarantees.

## 2 RELATED WORK

**Analysis of pairwise SGD via algorithmic stability.** Algorithmic stability has gained much attention in statistical learning theory due to its attractive properties, i.e., the independence to hypothesis function space and wide applicability (Lei & Ying, 2020; Hardt et al., 2016). This analysis technique is applied to investigate theoretical foundations of pairwise SGD (Lei et al., 2021b; 2020; Shen et al., 2019; Yang et al., 2021). For the convex pairwise SGD, Shen et al. (2019) established the bounds of the expected optimization error and excess risk after illustrating the trade-off between the stability and optimization error. Moreover, some systematic studies are provided in Lei et al. (2021b; 2020); Yang et al. (2021) to cover more general cases (i.e., without the bounded loss assumption or smoothness assumption). For the non-convex pairwise SGD, Lei et al. (2021b) investigated the stability and generalization of pairwise SGD under the gradient dominance condition, while the derived bounds

are not tight enough. Therefore, it is necessary to further explore learning guarantees of non-convex pairwise SGD from the perspective of algorithmic stability. Please see *Appendix D* for the outlines of algorithmic stability.

**Analysis of SGD with heavy-tailed gradient noise.** The heavy-tailed performance of SGD has been studied extensively, see e.g., Simsekli et al. (2019a;b); Nguyen et al. (2019); Hodgkinson & Mahoney (2021); Panigrahi et al. (2019). In a seminal paper, Vladimirova et al. (2019) found that the Bayesian neural network presents a heavier-tailed unit distribution than Gaussian prior (de G. Matthews et al., 2018; Lee et al., 2018) while deepening the model. After that, several works (Simsekli et al., 2019a;b; Panigrahi et al., 2019) verified that SGD also has heavier-tailed performance than sub-Gaussian distribution. It is demonstrated in Nguyen et al. (2019); Hodgkinson & Mahoney (2021) that the generalization ability of SGD may suffer from the heavy-tailed gradient noise. Based on uniform convergence analysis, the high probability guarantees for non-convex pointwise SGD are stated in Madden et al. (2020); Li & Liu (2022) under heavy-tailed gradient noise assumption. However, as far as we know, there are no stability-based learning guarantees for pairwise SGD with heavy tails. In this paper, we aim to make an effort to fill this theoretical gap.

## 3 PRELIMINARIES

This section provides the essential notations, definitions and assumptions, which lay the foundations for the subsequent analysis. Detail descriptions of notations are summarized in *Appendix A*.

### 3.1 NOTATIONS

For a sample space $\mathcal{Z}$, we assume that it contains an input space $\mathcal{X}$ and an output space $\mathcal{Y}$, i.e., $\mathcal{Z} = \mathcal{X} \times \mathcal{Y}$. According to an unknown probability measure $\rho$ defined on $\mathcal{Z}$, we draw each sample $z_i (1 \le i \le n)$ independently and get the training set $S := \{z_1, ..., z_n\} \in \mathcal{Z}^n$. The goal of pairwise learning is to find a data-driven predictor such that the population risk

$$F(w) := \mathbb{E}_{z,\tilde{z}}[f(w; z, \tilde{z})] \tag{1}$$

is as small as possible, where $f(w; z, \tilde{z}) : \mathcal{W} \times \mathcal{Z} \times \mathcal{Z} \to \mathbb{R}$ is a loss function, $w$ is the model parameter belonging to the hypothesis space $\mathcal{W}$, and $\mathbb{E}_{z,\tilde{z}}$ denotes the conditional expectation with respect to (w.r.t.) samples $z$ and $\tilde{z}$. Due to the inaccessibility of $F(w)$, we often formulate pairwise learning algorithms by minimizing the empirical risk

$$F_S(w) := \frac{1}{n(n-1)} \sum_{i,j \in [n], i \ne j} f(w; z_i, z_j), \ \ [n] := \{1, ..., n\}. \tag{2}$$

For feasibility, let $A(S)$ be the model parameter trained by algorithm $A : \mathcal{Z}^n \to \mathcal{W}$ on dataset $S$, and let

$$w(S) \in \arg\min_{w \in \mathcal{W}} F_S(w), \ \ w^* \in \arg\min_{w \in \mathcal{W}} F(w), \tag{3}$$

where $F(w), F_S(w)$ are defined in (1) and (2), respectively. Since $|\mathbb{E}_S[F_S(w(S))] - F(w^*)| = 0$, the excess risk of $A(S)$ can be decomposed by

$$|\mathbb{E}_S[F(A(S)) - F(w^*)]| \le |\mathbb{E}_S[F(A(S)) - F_S(A(S))]| + |\mathbb{E}_S[F_S(A(S)) - F_S(w(S))]|, \tag{4}$$

where $\mathbb{E}[\cdot]$ denotes the expectation w.r.t. all randomness and $w(S), w^*$ are defined in (3). Usually, we call the first term $|\mathbb{E}[F(A(S)) - F_S(A(S))]|$ as the generalization error and the second term $|\mathbb{E}[F_S(A(S)) - F_S(w(S))]|$ as the optimization error. This paper focuses on the generalization and optimization error estimates of the pairwise SGD with non-convex losses.

### 3.2 DEFINITIONS

We now introduce the definitions of SGD, minibatch SGD, on-average model stability and sub-Weibull random variable.

**Definition 3.1.** *(SGD for Pairwise Learning) For $t \in \mathbb{N}$, let $\{w_t\}$ be an update sequence of model parameters with the initial state $w_1 = 0$ and let $\{\eta_t\}$ be a stepsize sequence. Denote $\nabla f(w_t; z_{i_t}, z_{j_t})$ as the gradient of the loss function $f(w_t; z_{i_t}, z_{j_t})$ w.r.t. the first argument $w_t$, where $(z_{i_t}, z_{j_t})$ is a*

*sample pair selected to update model parameters in the $t$-th iteration, and $(i_t, j_t)$ is independently drawn from $\{(i, j) : i, j \in [n], i \neq j\}$. Then, the pairwise SGD is updated by*

$$w_{t+1} = w_t - \eta_t \nabla f(w_t; z_{i_t}, z_{j_t}). \tag{5}$$

**Remark 3.2.** *Different from the pointwise SGD (Lei & Ying, 2020; Hardt et al., 2016), Definition 3.1 involves dependent $\mathcal{O}(n^2)$ terms, which results in the additional barrier for stability analysis. To circumvent this barrier, a new concept of pairwise $\ell_1$ on-average model stability is proposed in Definition 3.5.*

**Definition 3.3.** *(Minibatch SGD for Pairwise Learning) For $\{w_t\}, \{\eta_t\}$ described in Definition 3.1 and the batch size $b$, denote $\nabla f(w_t; z_{i_{t,m}}, z_{j_{t,m}})$ as the gradient of the loss function $f(w_t; z_{i_{t,m}}, z_{j_{t,m}})$ w.r.t. the first argument $w_t$, where $m \in [b]$, $(z_{i_{t,m}}, z_{j_{t,m}})$ is the $m$-th sample pair selected to update model parameters in the $t$-th iteration, and $(i_{t,m}, j_{t,m})$ is independently drawn from $\{(i, j) : i, j \in [n], i \neq j\}$. Then, the pairwise minibatch SGD updates $\{w_t\}$ by*

$$w_{t+1} = w_t - \frac{\eta_t}{b} \sum_{m=1}^{b} \nabla f(w_t; z_{i_{t,m}}, z_{j_{t,m}}). \tag{6}$$

**Remark 3.4.** *The pairwise minibatch SGD in Definition 3.3 reduces to the pairwise SGD in Definition 3.1 as $b = 1$. Note that, when $b = n(n-1)$, Definition 3.3 is inconsistent with the full-batch SGD for the reason that $(z_{i_{t,m}}, z_{j_{t,m}})$ is independently selected from all sample pairs $\{(z_i, z_j) : z_i, z_j \in S, z_i \neq z_j\}$, which means that some certain sample pair can be selected more than once at each itration.*

**Definition 3.5.** *Let $S = \{z_i\}_{i=1}^{n}$, $S' = \{z_i'\}_{i=1}^{n}$ be drawn independently from $\rho$. Define*

$$S_{i,j} = \{z_1, ..., z_{i-1}, z_i', z_{i+1}, ..., z_{j-1}, z_j', z_{j+1}, ..., z_n\}, \ \ \forall i, j \in [n], i \neq j.$$

*Denote $\|\cdot\|$ as the Euclidean norm. A pairwise learning algorithm $A$ is $\ell_1$ on-average model $\epsilon$-stable if*

$$\mathbb{E}_{S,S',A} \left[ \frac{1}{n(n-1)} \sum_{i,j \in [n], i \neq j} \|A(S_{i,j}) - A(S)\| \right] \leq \epsilon.$$

The pointwise $\ell_1$ on-average model stability, proposed by Lei & Ying (2020), has shown the powerful ability for generalization analysis (Lei et al., 2021b; 2020), which is milder than the uniform model (argument) stability (Liu et al., 2017). Motivated by the on-average stability of Lei et al. (2020), Definition 3.5 nails down the pairwise $\ell_1$ on-average model stability. Note that, Lei et al. (2020) mainly considers $\ell_2$ on-average model stability instead of $\ell_1$ on-average model stability.

Indeed, the on-average model stability is used to measure the model parameter sensitivity to a small perturbation of $S$, which is different from the ones concerning the sensitivity of loss function value, e.g., the uniform stability (Hardt et al., 2016; Bousquet & Elisseeff, 2002; Shalev-Shwartz et al., 2010) and the on-average stability (Lei et al., 2020; Kuzborskij & Lampert, 2018; Lei & Ying, 2021).

**Definition 3.6.** *(Vladimirova et al., 2020) We say $X$ is a sub-Weibull random variable if the moment generating function (MGF) $\mathbb{E}\left[\exp\left((|X|/K)^{\frac{1}{\theta}}\right)\right] \leq 2$ for some positive parameters $K$ and $\theta \geq 1/2$, and denote it as $X \sim subW(\theta, K)$.*

The sub-Weibull random variable $X$ becomes the sub-Gaussian as $\theta = 1/2$ (Vershynin, 2018) or the sub-Exponential distribution as $\theta = 1$ (Vladimirova et al., 2020). We concern the pairwise SGD with heavy tails and let $\theta > 1/2$ in the rest of this paper. For ease of understanding, some necessary preliminaries of sub-Weibull distribution are provided in *Appendix B*.

## 3.3 ASSUMPTIONS

We first describe two common assumptions, namely Lipschitz continuity and smoothness.

**Assumption 3.7.** *(a) For any $z, \tilde{z} \in \mathcal{Z}$, $w, w' \in \mathcal{W}$ and $L > 0$, a differentiable loss function $f(w; z, z')$ is $L$-Lipschitz continuous w.r.t the first argument $w$ if $\|\nabla f(w; z, \tilde{z})\| \leq L$, which means that $|f(w; z, \tilde{z}) - f(w'; z, \tilde{z})| \leq L\|w - w'\|$.*

*(b) For any $z, \tilde{z} \in \mathcal{Z}$, $w, w' \in \mathcal{W}$ and $\beta > 0$, a differentiable loss function $f(w; z, z')$ is $\beta$-smooth w.r.t the first argument $w$ if $\|\nabla f(w; z, \tilde{z}) - \nabla f(w'; z, \tilde{z})\| \leq \beta\|w - w'\|$.*

Some previous work assumed the gradient and the loss function itself are both Lipschitz (Hardt et al., 2016; Lei & Ying, 2020). However, the Lipschitz continuity assumption may be fragile since the parameter $L$ may be very large or even infinite for some learning environments (Lei & Ying, 2020). In these cases, many stability-based generalization bounds under this assumption don't match the algorithmic deployment in real applications. Hence, we introduce the assumption of the heavy-tailed gradient noise below (Assumption 3.8) to remove the bounded gradient assumption in our analysis. As for smoothness, it is assumed throughout the whole paper.

**Assumption 3.8.** *(Sub-Weibull Gradient Noise) For the $t$-th iteration of (5), we assume* $\nabla f(w_t; z_{i_t}, z_{j_t}) - \nabla F_S(w_t) \sim subW(\theta, K)$ *with* $\theta > 1/2, K > 0$, *i.e.,*

$$\mathbb{E}_{i_t, j_t} \left[ \exp \left( \left( \frac{\|\nabla f(w_t; z_{i_t}, z_{j_t}) - \nabla F_S(w_t)\|}{K} \right)^{\frac{1}{\theta}} \right) \right] \leq 2.$$

Recently, rich works have shown that SGD and its variants exhibit heavier noise than sub-Gaussian (Simsekli et al., 2019a;b; Madden et al., 2020; Panigrahi et al., 2019; Zhang et al., 2020a; Wang et al., 2021). Hence, it is natural to consider Assumption 3.8 here for the pairwise SGD in Definition 3.1 with heavy tails. In our analysis, the gradient noise assumption provides some refined bounds of gradient noise (Lemma C.3 in *Appendix C.1*) which are key to bridging the connection between $\ell_1$ on-average model stability and generalization error (Theorem 4.1 (b)) and stating stability bounds (Theorems 4.4, 4.6, 4.9) without the bounded gradient assumption.

**Assumption 3.9.** *(Polyak-Lojasiewicz (PL) condition) For any $w \in \mathcal{W}$ and $S \in \mathcal{Z}^n$, the empirical risk $F_S(w)$ (2) satisfies the PL condition with parameter $\mu > 0$ if*

$$\|\nabla F_S(w)\|^2 \geq 2\mu \left( F_S(w) - F_S(w(S)) \right).$$

The PL condition, also called gradient dominance condition (Lei et al., 2021b; Zhou et al., 2022; Foster et al., 2018; Reddi et al., 2016), can be viewed as a mild control over the curvature of loss function and has been employed for the non-convex generalization analysis (Lei & Tang, 2021; Li & Liu, 2022; Lei & Ying, 2021). This condition demonstrates that the lower bound of the quadratic of objective gradient is $2\mu \left( F_S(w) - F_S(w(S)) \right)$ and will increase as the model parameter $w$ is far away from the empirically optimal parameter $w(S)$ (Karimi et al., 2016). Note that, the PL condition assures that any $w$ satisfying $\|\nabla F_S(w)\| = 0$ is a global minimizer (Charles & Papailiopoulos, 2018).

## 4 MAIN RESULTS

This section builds the quantitative relationships between $\ell_1$ on-average model stability and generalization error firstly, which is the basis of our theoretical analysis. Then, we present the error bound for the general non-convex pairwise SGD in Section 4.1 and its refined version in Section 4.2. Section 4.3 further considers the approximation performance of SGD (5) under the PL condition (Assumption 3.9) and Section 4.4 extends the related results to the case of pairwise minibatch SGD. We provide the proof sketch of our results in Appendix C and summarize the comparisons of related results in Tables 1, 2, and 4 (*Appendix C.7*). All detailed proofs are provided in *Appendix C*. Note that, all bounds in the main text are in expectation. They can be developed to establish high probability bounds which are provided in Appendix C.8.

**Theorem 4.1.** *Let $S, S'$ and $S_{i,j}$ be constructed as Definition 3.5.*

*(a) Assume that pairwise learning algorithm A, associated with L-Lipschitz continuous loss function, is $\ell_1$ on-average model $\epsilon$-stable. Then,*

$$|\mathbb{E}[F_S(A(S)) - F(A(S))]| \leq L\epsilon.$$

*(b) Assume that pairwise SGD A, associated with loss function whose gradient noise obeys $subW(\theta, K)$, is $\ell_1$ on-average model $\epsilon$-stable. Then,*

$$|\mathbb{E}[F(A(S)) - F_S(A(S))]| \leq 2\mathbb{E}[F_S(A(S))] + (4\theta)^{\theta} K\epsilon.$$

Theorem 4.1(a) shows the generalization error in expectation can be controlled by the $\ell_1$ on-average model stability bound even for the general non-convex pairwise SGD. Theorem 4.1 (b) verifies the

generalization bound via $\ell_1$ on-average model stability enjoys some attractive properties, e.g., independence of the Lipschitz continuity assumption. Additionally, we can easily find the generalization error monotonically increasing with the expected empirical risk $\mathbb{E}[F_S(A(S))]$. This means the optimizer minimizing $F_S(A(S))$ contributes to the improvement of generalization ability (Lei & Ying, 2020). The common choices $\mathbb{E}[F_S(A(S))] = \mathcal{O}\left(n^{-1}\right)$ can be made to obtain some satisfactory bounds as shown in Corollaries 4.5, 4.7 and 4.10.

Theorem 4.1 (a) is consistent with the related results of stability and generalization for pointwise learning (Theorem 2 in Lei & Ying (2020)) and pairwise learning (Theorem 1 in Lei et al. (2021b)), where the slight difference is induced by the divergence among stability definitions. However, the Lipschitz continuity condition is necessary for our proof framework under the case without heavy-tailed gradient noise. Theorem 4.1 (b) is a novel quantitative relationship between $\ell_1$ on-average model stability and generalization error under heavy-tailed gradient noise assumption.

Table 1: Summary of stability bounds for pairwise SGD with non-convex loss functions ($\sqrt{}$-has such a property; $\times$-hasn't such a property; $L, \beta, \mu, \theta$-the parameters of Lipschitz continuity, smoothness, PL condition and sub-Weibull distribution; $c$-a non-negative constant; $b' = \frac{b}{2n(n-1)\left(1+\frac{b-1}{n(n-1)}\right)}$). See *Appendix D* for details of stability tools.

| Reference | Assumptions | | | Stability Tool | Stability Bound |
|---|---|---|---|---|---|
| | $L$ | $\mu$ | $\theta$ | | |
| Shen et al. (2019) (Thm. 3.5) | $\sqrt{}$ | $\times$ | $\times$ | Uniform stability | $\mathcal{O}\left((\beta n)^{-1}L^{\frac{2}{\beta c+1}}T^{\frac{\beta c}{\beta c+1}}\right)$ |
| Lei et al. (2021b) (Thm. 15) | $\sqrt{}$ | $\sqrt{}$ | $\times$ | Uniform stability | $\mathcal{O}\left((\beta n)^{-1}L^2 T^{\frac{\beta c}{\beta c+1}}\right)$ |
| Ours (Thm. 4.2) | $\sqrt{}$ | $\times$ | $\times$ | On-average model stability | $\mathcal{O}\left((\beta n)^{-1}LT^{\frac{1}{2}}\log T\right)$ |
| Ours (Thm. 4.4) | $\times$ | $\times$ | $\sqrt{}$ | On-average model stability | $\mathcal{O}\left((\beta n)^{-1}(\Gamma(2\theta+1))^{\frac{1}{2}}T^{\frac{1}{2}}(\log T)^{\frac{3}{2}}\right)$ |
| Ours (Thm. 4.6) | $\times$ | $\sqrt{}$ | $\sqrt{}$ | On-average model stability | $\mathcal{O}\left((\beta n)^{-1}(\Gamma(2\theta+1))^{\frac{1}{2}}T^{\frac{1}{4}}(\log T)^{\frac{3}{2}}\right)$ |
| Ours (Thm. 4.9) | $\times$ | $\sqrt{}$ | $\sqrt{}$ | On-average model stability | $\mathcal{O}\left((\beta n)^{-1}(\Gamma(2\theta+1))^{\frac{1}{2}}T^{b'}(\log T)^{\frac{3}{2}}\right)$ |

## 4.1 GENERAL NON-CONVEX PAIRWISE SGD

Now we state the quantitative characterization of on-average model stability for the general non-convex pairwise SGD.

**Theorem 4.2.** *Given $S, S'$ and $S_{i,j}$ in Definition 3.5, let $\{w_t\}$ and $\{w'_t\}$ be produced by (5) on $S$ and $S_{i,j}$ respectively, where $\eta_t = \eta_1 t^{-1}, \eta_1 \leq (2\beta)^{-1}$, and let the parameters $A(S) = w_T$ and $A(S_{i,j}) = w'_T$ after $T$ iterations. Assume that the loss function $f(w; z, z')$ is $L$-Lipschitz and $\beta$-smooth w.r.t. the first argument $w$. Then, there holds*

$$\frac{1}{n(n-1)} \sum_{i,j\in[n],i\neq j} \mathbb{E}\left[\|w_T - w'_T\|\right] \leq \mathcal{O}\left((\beta n)^{-1}LT^{\frac{1}{2}}\log T\right).$$

Theorem 4.2 illustrates $\ell_1$ on-average model stability bound $\mathcal{O}\left((\beta n)^{-1}LT^{\frac{1}{2}}\log T\right)$ for non-convex pairwise SGD when the loss function is Lipschitz continuous and smooth. Shen et al. (2019) provided a uniform stability bound $\mathcal{O}\left((\beta n)^{-1}L^{\frac{2}{\beta c+1}}T^{\frac{\beta c}{\beta c+1}}\right)$, where the constant $c > 0$. Thus, the bound of Theorem 4.2 is tighter than it when $0 < c \leq \left(1/\log_{L^2 T^{-1}}(LT^{-1/2}\log T) - 1\right)/\beta$. Lei et al. (2021b) provided a uniform stability bound $\mathcal{O}\left((\beta n)^{-1}L^2 T^{\frac{\beta c}{\beta c+1}}\right)$, where the constant $c = 1/\mu$ ($\mu$ is the parameter of PL condition). In general, $\mu$ is typically a very small value (Examples 1 and 2 in Lei & Ying (2021)) which leads to a large value of $c$. Thus, $T^{\frac{\beta c}{\beta c+1}}$ is closer to $T$ than $(T^{1/2}\log T)$ in our bound. In other words, our bound is tighter than Lei et al. (2021b).

Combining Theorem 4.1 (a) and Theorem 4.2 yields the following generalization bound.

**Corollary 4.3.** *Under Assumptions 3.7 (a) and 3.7 (b), for the pairwise SGD (5) with $T$ iterations,*

$$|\mathbb{E}[F(w_T) - F_S(w_T)]| \leq \mathcal{O}\big((\beta n)^{-1} L^2 T^{\frac{1}{2}} \log T\big).$$

In terms of uniform stability analysis, Shen et al. (2019) provided the generalization error bound $\mathcal{O}\left((\beta n)^{-1} L^{\frac{2}{\beta c+1}} T^{\frac{\beta c}{\beta c+1}}\right)$ for non-convex pairwise SGD. When $\log T \leq L^{\frac{-2\beta c}{\beta c+1}} T^{\frac{\beta c-1}{2\beta c+2}}$, the derived result in Corollary 4.3 is better than Shen et al. (2019).

### 4.2 Non-convex Pairwise SGD without Lipschitz Condition

Now we state the refined bounds of stability and generalization error by leveraging the heavy-tailed gradient noise assumption to remove the Lipschitz continuity assumption.

**Theorem 4.4.** *Given $S, S'$ and $S_{i,j}$ described in Definition 3.5, let $\{w_t\}$ and $\{w'_t\}$ be produced by (5) on $S$ and $S_{i,j}$ respectively, where $\eta_t = \eta_1 t^{-1}, \eta_1 \leq (2\beta)^{-1}$, and let the parameters $A(S) = w_T$ and $A(S_{i,j}) = w'_T$ after $T$ iterations. Assume that the loss function $f(w; z, z')$ is $\beta$-smooth. Take $\Gamma(x) = \int_0^\infty t^{x-1} e^{-t} dt$. Under Assumption 3.8, there holds*

$$\frac{1}{n(n-1)} \sum_{i,j \in [n], i \neq j} \mathbb{E}\left[\|w_T - w'_T\|\right] \leq \mathcal{O}\big((\beta n)^{-1} (\Gamma(2\theta + 1))^{\frac{1}{2}} T^{\frac{1}{2}} (\log T)^{\frac{3}{2}}\big).$$

Theorem 4.4 assures the upper bound with the order $\mathcal{O}\big((\beta n)^{-1} (\Gamma(2\theta + 1))^{\frac{1}{2}} T^{\frac{1}{2}} (\log T)^{\frac{3}{2}}\big)$ for $\ell_1$ on-average model stability, where the heavy-tailed gradient noise assumption is employed to get rid of the bounded gradient assumption. Observe that, the dependence on $T$ for the bound of Theorem 4.4 is just $\sqrt{\log T}$-times larger than Theorem 4.2 and the additional dependence on the heavy tail parameter $\theta$ is often bounded (Vladimirova et al., 2020). Thus, it is better than the dependence on the Lipschitz parameter $L$ which is likely infinite for some learning environments. Due to the above reasons, the bound of Theorem 4.4 is tighter than the one of Theorem 4.2.

Theorem 3 in Lei et al. (2021b) provided a $\ell_2$ on-average model stability bound $\mathcal{O}\left(n^{-1}\left(1 + \frac{T}{n}\right) \sum_{t=1}^T \eta_t^2 \mathbb{E}[F_S(w_t)]\right)$ for pairwise SGD with convex loss functions, which involves $\sum_{t=1}^T \eta_t^2 \mathbb{E}[F_S(w_t)]$. However, it is hard to ensure the summation of empirical risks is small enough. Therefore, the current result enjoys much adaptivity and flexibility since it is not affected by the quality of the initial empirical risks and nears optimum under the milder assumptions, i.e., non-convex loss and heavy-tailed gradient noise.

**Corollary 4.5.** *Under Assumptions 3.7 (b) and 3.8, for the pairwise SGD (5) with $T$ iterations,*

$$|\mathbb{E}[F(w_T) - F_S(w_T)]| \leq \mathcal{O}\big((\beta n)^{-1} (4\theta)^\theta (\Gamma(2\theta + 1))^{\frac{1}{2}} T^{\frac{1}{2}} (\log T)^{\frac{3}{2}} + \mathbb{E}[F_S(w_T)]\big).$$

The generalization bound in Corollary 4.5 is obtained by combining Theorem 4.1 (b) and Theorem 4.4. If $\mathbb{E}[F_S(w_T)] = \mathcal{O}\left(n^{-1}\right)$, we can derive $\mathbb{E}[F(w_T) - F_S(w_T)] = \mathcal{O}\big((\beta n)^{-1} (4\theta)^\theta (\Gamma(2\theta + 1))^{\frac{1}{2}} T^{\frac{1}{2}} (\log T)^{\frac{3}{2}}\big)$, where $(4\theta)^\theta$ is smaller than the bounded dependence $\Gamma(2\theta + 1)$ on heavy-tailed parameter $\theta$ (Li & Liu, 2022).

### 4.3 Non-convex Pairwise SGD with PL Condition

Inspired from Li & Liu (2022), we further investigate learning guarantees of non-convex pairwise SGD under the PL condition which assures that the estimator $w$ satisfying $\|\nabla F_S(w)\| = 0$ is a global minimizer of empirical risk $F_S(w)$ (2).

**Theorem 4.6.** *Given $S, S'$ and $S_{i,j}$ in Definition 3.5 and $w(S)$ in (3), let $\{w_t\}$ and $\{w'_t\}$ be produced by (5) on $S$ and $S_{i,j}$ respectively, where $\eta_t = \eta_1 t^{-1}, \eta_1 \leq (4\beta)^{-1}, 1 - \mu\eta_1 \geq 0$. Assume that the loss function $f(w; z, z')$ is $\beta$-smooth w.r.t. the first argument $w$. Take $a_1 = 1 - \prod_{i=1}^t \left(1 - \frac{1}{2}\mu\eta_i\right)$. Under Assumptions 3.8 and 3.9, there holds*

$$\frac{1}{n(n-1)} \sum_{i,j \in [n], i \neq j} \mathbb{E}\left[\|w_T - w'_T\|\right] \leq \mathcal{O}\left(\frac{1}{n\sqrt{\beta}} T^{\frac{1}{4}} \log T \sqrt{a_1 \mathbb{E}[F_S(w(S))] + \frac{\Gamma(2\theta + 1) \log T}{\beta}}\right).$$

Table 2: Summary of excess risk bounds for non-convex SGD via uniform convergence approaches and stability analysis (♣-uniform convergence; ♠-stability; ▼-pointwise learning; ▲-pairwise learning; $\beta, \mu, \theta$-the parameters of smoothness, PL condition and sub-Weibull distribution; $\surd$-has such a property; $\times$-hasn't such a property; $d$-dimension of hypothesis function space; $*$-high-probability bound; $\delta$-some probability; $b' = \frac{b}{2n(n-1)\left(1+\frac{b-1}{n(n-1)}\right)}$).

| Reference | Assumptions | | | Excess risk bound |
|---|---|---|---|---|
| | $\beta$ | $\mu$ | $\theta$ | |
| Madden et al. (2020) ♣▼ (Thm. 9) | $\surd$ | $\surd$ | $\times$ | $*\mathcal{O}\left(T^{-1}\log\left(\frac{1}{\delta}\right)\right)$ |
| Lei & Tang (2021) ♣▼ (Thm. 7) | $\surd$ | $\surd$ | $\times$ | $*\mathcal{O}\left(n^{-1}\left(d+\log\left(\frac{1}{\delta}\right)\right)\log^2 n \log^2\left(\frac{1}{\delta}\right)\right)$ |
| Li & Liu (2022) ♣▼ (Thm. 3.11) | $\surd$ | $\surd$ | $\surd$ | $*\mathcal{O}\left(n^{-1}\left(d+\log\left(\frac{1}{\delta}\right)\right)\log^{2\theta+1}\left(\frac{1}{\delta}\right)\log^{\frac{3(\theta-1)}{2}}\left(\frac{n}{\theta}\right)\log n\right)$ |
| Lei et al. (2021b) ♠▲ (Thm. 15) | $\surd$ | $\surd$ | $\times$ | $\mathcal{O}\left((\beta n)^{-1}L^2 T^{\frac{\beta}{\beta+\mu}}+T^{-1}\right)$ |
| Ours ♠▲ (Thm. 4.8) | $\surd$ | $\surd$ | $\surd$ | $\mathcal{O}\left(\frac{\Gamma(2\theta+1)}{\beta T}+(\beta n)^{-1}T^{\frac{1}{4}}(4\theta)^{\theta}(\Gamma(2\theta+1))^{\frac{1}{2}}(\log T)^{\frac{3}{2}}\right)$ |
| Ours ♠▲ (Thm. 4.11) | $\surd$ | $\surd$ | $\surd$ | $\mathcal{O}\left(\frac{\Gamma(2\theta+1)}{\beta T}+(\beta n)^{-1}T^{b'}(4\theta)^{\theta}(\Gamma(2\theta+1))^{\frac{1}{2}}(\log T)^{\frac{3}{2}}\right)$ |

Compared with Theorem 4.4, the stability bound $\mathcal{O}\left((\beta n)^{-1}(\Gamma(2\theta+1))^{\frac{1}{2}}T^{\frac{1}{4}}(\log T)^{\frac{3}{2}}\right)$ in Theorem 4.6 involves a different term $T^{\frac{1}{4}}$ which is better than $T^{\frac{1}{2}}$ when $\mathbb{E}[F_S(w(S))]=\mathcal{O}\left(n^{-1}\right)$.

The following generalization bound is derived by integrating Theorem 4.1 (b) and Theorem 4.6.

**Corollary 4.7.** *Under Assumptions 3.7 (b)-3.9, for the pairwise SGD (5) with $T$ iterations,*

$$|\mathbb{E}[F(w_T)-F_S(w_T)]|$$
$$\leq \mathcal{O}\left(\mathbb{E}[F_S(w_T)]+n^{-1}\beta^{-\frac{1}{2}}T^{\frac{1}{4}}(4\theta)^{\theta}\log T\sqrt{a_1\mathbb{E}[F_S(w(S))]+\beta^{-1}\Gamma(2\theta+1)\log T}\right).$$

Corollary 4.7 states the generalization bound $\mathcal{O}((\beta n)^{-1}(4\theta)^{\theta}(\Gamma(2\theta+1))^{\frac{1}{2}}T^{\frac{1}{4}}(\log T)^{\frac{3}{2}})$ when $\mathbb{E}[F_S(w(S))]\leq \mathbb{E}[F_S(w_T)]=\mathcal{O}\left(n^{-1}\right)$. Lei et al. (2021b) have established the first generalization bound $\mathcal{O}\left((\beta n)^{-1}L^2 T^{\beta/(\beta+\mu)}\right)$ for non-convex pairwise SGD under the gradient dominance condition. Different from the existing work that relies on uniform stability (Lei et al., 2021b), Corollary 4.7 provides tighter generalization bound under weaker conditions when $(4\theta)^{\theta}(\Gamma(2\theta+1))^{\frac{1}{2}}(\log T)^{\frac{3}{2}}\leq L^2 T^{\frac{\beta}{\beta+\mu}-\frac{1}{4}}$.

**Theorem 4.8.** *Given $w^*$ in (3) and $\{w_t\}$ produced by (5) on $S$, where $\eta_t=\eta_1 t^{-1}, \eta_1\leq(4\beta)^{-1}, 1-\mu\eta_1\geq 0$. Assume that the function $f(w;z,z')$ is $\beta$-smooth. Under Assumptions 3.8 and 3.9, after $T$ iterations, there hold $\mathbb{E}[F_S(w_T)-F_S(w(S))]\leq\mathcal{O}\left(T^{-2}+(\beta T)^{-1}\Gamma(2\theta+1)\right)$ and*

$$|\mathbb{E}[F(w_T)-F(w^*)]|\leq\mathcal{O}\left((\beta T)^{-1}\Gamma(2\theta+1)+(\beta n)^{-1}T^{\frac{1}{4}}(4\theta)^{\theta}(\Gamma(2\theta+1))^{\frac{1}{2}}(\log T)^{\frac{3}{2}}\right).$$

With the help of the PL condition, we can guarantee that the algorithm can find a global minimizer. Therefore, we can use $\mathbb{E}[F_S(w_T)-F_S(w(S))]$ instead of $\|\nabla F_S(w_T)\|$ to measure the optimization performance of pairwise SGD in Definition 3.1. For the non-convex pairwise SGD, Theorem 4.8 provides the optimization error bound $\mathcal{O}\left((\beta T)^{-1}\Gamma(2\theta+1)\right)$ and the excess risk bound $\mathcal{O}\left((\beta T)^{-1}\Gamma(2\theta+1)+(\beta n)^{-1}T^{\frac{1}{4}}(4\theta)^{\theta}(\Gamma(2\theta+1))^{\frac{1}{2}}(\log T)^{\frac{3}{2}}\right)$ by (4). The derived optimization error bound is comparable with $\mathcal{O}\left(T^{-1}\right)$ stated in Lemma D.1 (e) of *Appendix D* in Lei et al. (2021b). For excess risk, our bound is $\mathcal{O}\left(n^{-\frac{3}{4}}\beta^{-1}(4\theta)^{\theta}(\Gamma(2\theta+1))^{\frac{1}{2}}(\log n)^{\frac{3}{2}}\right)$ as taking $T\asymp n$, which is comparable with the related results (Lei & Tang, 2021; Li & Liu, 2022) for the pointwise SGD and enjoys nice property, i.e., the independence of the dimension $d$.

Moreover, the excess risk bound $\mathcal{O}\left((\beta n)^{-1}L^2 T^{\frac{\beta}{\beta+\mu}}+T^{-1}\right)$ (Lei et al., 2021b) implies the convergence order $\mathcal{O}\left(n^{-\frac{\beta/\mu+1}{2\beta/\mu+1}}\right)$ as $T\asymp n^{\frac{\beta+\mu}{2\beta+\mu}}$, which is often slower than ours with $T\asymp n$. As

shown in Table 2, our results fill the theoretical gap of stability-based excess risk analysis for the non-convex pairwise SGD with heavy tails, and guarantee a satisfactory convergence rate.

## 4.4 NON-CONVEX MINIBATCH SGD

We further investigate the stability and generalization of non-convex pairwise minibatch SGD. To our surprise, this issue has not been studied in machine learning literature before.

**Theorem 4.9.** *Given $S, S'$ and $S_{i,j}$ in Definition 3.5, let $\{w_t\}$ and $\{w'_t\}$ be produced by the pairwise minibatch SGD (6) (batchsize $b$) on $S$ and $S_{i,j}$ respectively, where $\eta_t = \eta_1 t^{-1}, \eta_1 \leq \frac{b}{2n(n-1)\left(1+\frac{b-1}{n(n-1)}\right)\beta} = b'/\beta, 1 - \mu\eta_1 \geq 0$, and let the parameters $A(S) = w_T$ and $A(S_{i,j}) = w'_T$ after $T$ iterations. Assume that $f(w; z, z')$ is $\beta$-smooth w.r.t. the first argument $w$. Under Assumptions 3.8, 3.9, there holds*

$$\frac{1}{n(n-1)} \sum_{i,j\in[n],i\neq j} \mathbb{E}\|w_T - w'_T\| \leq \mathcal{O}\left(n^{-1}\beta^{-\frac{1}{2}}T^{b'}\log T\sqrt{a_1\mathbb{E}[F_S(w(S))] + \frac{\Gamma(2\theta+1)\log T}{\beta}}\right).$$

The main challenge of proving Theorem 4.9 is to tackle the sampling procedures of minibatch SGD. This technique barrier is surmounted by introducing the binomial distribution to reformulate (6) in *Appendix C.6*. To the best of our knowledge, Theorem 4.9 provides the first near-optimal stability bound $\mathcal{O}\left((\beta n)^{-1}(\Gamma(2\theta+1))^{\frac{1}{2}}T^{b'}(\log T)^{\frac{3}{2}}\right)$ for the minibatch case when $\mathbb{E}[F_S(w(S))] \leq \mathcal{O}\left(n^{-1}\right)$. If $b = 1$, the order of this upper bound is $\mathcal{O}\left((\beta n)^{-1}(\Gamma(2\theta+1))^{\frac{1}{2}}T^{\frac{1}{2n(n-1)}}(\log T)^{\frac{3}{2}}\right) \leq \mathcal{O}\left((\beta n)^{-1}(\Gamma(2\theta+1))^{\frac{1}{2}}T^{\frac{1}{4}}(\log T)^{\frac{3}{2}}\right)$, which recovers the result of Theorem 4.6.

Combining the above stability bound with Theorem 4.1 (b), we derive the following generalization bound for the minibatch SGD in Definition 3.3.

**Corollary 4.10.** *Under Assumptions 3.7 (b)-3.9, for the minibatch SGD (6) with $T$ iterations,*

$$|\mathbb{E}[F(w_T) - F_S(w_T)]|$$
$$\leq \mathcal{O}\left(\mathbb{E}[F_S(w_T)] + n^{-1}\beta^{-\frac{1}{2}}T^{b'}(4\theta)^\theta \log T\sqrt{a_1\mathbb{E}[F_S(w(S))] + \beta^{-1}\Gamma(2\theta+1)\log T}\right).$$

The generalization error bound $\mathcal{O}\left((\beta n)^{-1}T^{b'}(4\theta)^\theta(\Gamma(2\theta+1))^{\frac{1}{2}}(\log T)^{\frac{3}{2}}\right)$ is from the analysis of Theorem 4.9 when $\mathbb{E}[F_S(w(S))] \leq \mathbb{E}[F_S(w_T)] = \mathcal{O}\left(n^{-1}\right)$.

**Theorem 4.11.** *Let $A(S)$ be produced by the pairwise minibatch SGD (6) on $S$, where $\eta_t = \eta_1 t^{-1}, \eta_1 \leq b'/\beta, 1 - \mu\eta_1 \geq 0$. Assume that $f(w; z, z')$ is $\beta$-smooth w.r.t. the first argument $w$. Under Assumptions 3.8 and 3.9, after $T$ iterations, there hold $\mathbb{E}[F_S(w_T) - F_S(w(S))] \leq \mathcal{O}\left(T^{-2} + (\beta T)^{-1}\Gamma(2\theta+1)\right)$ and*

$$|\mathbb{E}[F(w_T) - F(w^*)]| \leq \mathcal{O}\left((\beta T)^{-1}\Gamma(2\theta+1) + (\beta n)^{-1}T^{b'}(4\theta)^\theta(\Gamma(2\theta+1))^{\frac{1}{2}}(\log T)^{\frac{3}{2}}\right).$$

Theorem 4.11 provides the first stability-based optimal optimization error and excess risk bounds for non-convex pairwise minibatch SGD with heavy tails and the PL condition. Compared with Theorem 4.8, the minibatch strategy damages the learning guarantee, which is consistent with previous empirical observations (Li et al., 2014; Lin et al., 2020). When $b \to n(n-1)$, the rate approximates $\mathcal{O}\left((\beta T)^{-1}\Gamma(2\theta+1) + (\beta n)^{-1}T^{\frac{1}{3}}(4\theta)^\theta(\Gamma(2\theta+1))^{\frac{1}{2}}(\log T)^{\frac{3}{2}}\right)$. Particularly, the detailed comparisons of our results are summarized in Table 4 of *Appendix C.7*.

## 5 CONCLUSIONS

This paper aims to fill the theoretical gap on the learning guarantees of non-convex pairwise SGD with heavy tails. We stated the first near-optimal bounds of generalization error and excess risk for the non-convex pairwise SGD respectively, where the technique of algorithmic stability analysis is developed to overcome the obstacle induced by the complicated pairwise objective and the minibatch strategy. As a natural extension of Li & Liu (2022), our results also verify the effect of the heavy-tailed gradient noise on removing the bounded gradient assumption of the pairwise loss function. In the future, it is interesting to further investigate the stability and generalization of pairwise minibatch SGD with other heavy-tailed distributions (such as $\alpha$-stable distributions (Simsekli et al., 2019b)).

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

# A    NOTATIONS

The main notations of this paper are summarized in Table 3.

Table 3: Summary of main notations involved in this paper.

| Notations | Descriptions |
|---|---|
| SGD | stochastic gradient descent |
| $\mathcal{Z}$ | the compact sample space associated with input space $\mathcal{X}$ and output space $\mathcal{Y}$ |
| $z$ | the random sample sampling from $\mathcal{Z}$ |
| $n$ | the numbers of samples sampling from $\mathcal{Z}$ |
| $b$ | the batch size |
| $w, \mathcal{W}$ | the model parameter and hypothesis function space, respectively |
| $S$ | the training dataset defined as $\{z_1, ..., z_n\} \in \mathcal{Z}^n$ |
| $d$ | the dimension of hypothesis function space |
| $f(w)$ | the pairwise loss function defined as $f(w; z, \tilde{z})$ |
| $\nabla f$ | the gradient of $f(w; z, \tilde{z})$ to the first argument $w$ |
| $F(w), F_S(w)$ | the population risk and empirical risk based on training dataset $S$, respectively |
| $T$ | the number of iterative steps for SGD |
| $w_T$ | the model parameter derived by SGD after $T$-th update |
| $w(S)$ | the optimal model based on the empirical risk, $w(S) = \arg \min\limits_{w \in \mathcal{W}} F_S(w)$ |
| $w^*$ | the optimal model based on the population risk, $w^* = \arg \min\limits_{w \in \mathcal{W}} F(w)$ |
| $\eta_T$ | the step size at the $T$-th update |
| $A, A(S)$ | the given algorithm and its output model parameter based on $S$ respectively |
| $L, \beta, \mu$ | the parameters of Lipschitz continuity, smoothness and PL condition respectively |
| $\theta, K$ | the parameters of sub-Weibull distribution |
| $\asymp$ | $n_+ \asymp n_-$ if there exist positive constants $c_1, c_2$ such that $c_1 n_+ \leq n_- \leq c_2 n_+$ |
| $[\cdot]$ | $[n] := \{1, ..., n\}$ |
| $e$ | the base of the natural logarithm |
| $\Gamma(x)$ | $\Gamma(x) = \int_0^\infty t^{x-1} e^{-t} dt$ |
| $\|\cdot\|$ | the Euclidean norm |
| $\delta$ | some probability |
| $\epsilon$ | the parameters of $\ell_1$ on-average model stability |
| $\ell_1$ | the type of on-average model stability |
| $A$ | a pairwise learning algorithm |
| $c$ | a non-negative constant |
| $v_S^2$ | the parameter of bounded variance assumption |

## B    PRELIMINARIES OF SUB-WEIBULL DISTRIBUTION

The original definition of sub-Weibull distribution (Vladimirova et al., 2020) is the class of distributions satisfying

$$\mathbb{P}(|X| \geq x) \leq a \exp -bx^{1/\theta}, \text{ for all } x \geq 0, \text{ for some } \theta, a, b > 0.$$

According to the original definition, we present some sub-Weibull survival curves with varying tail parameters $\theta$ in Figure 1 which is inspired by Vladimirova et al. (2020).

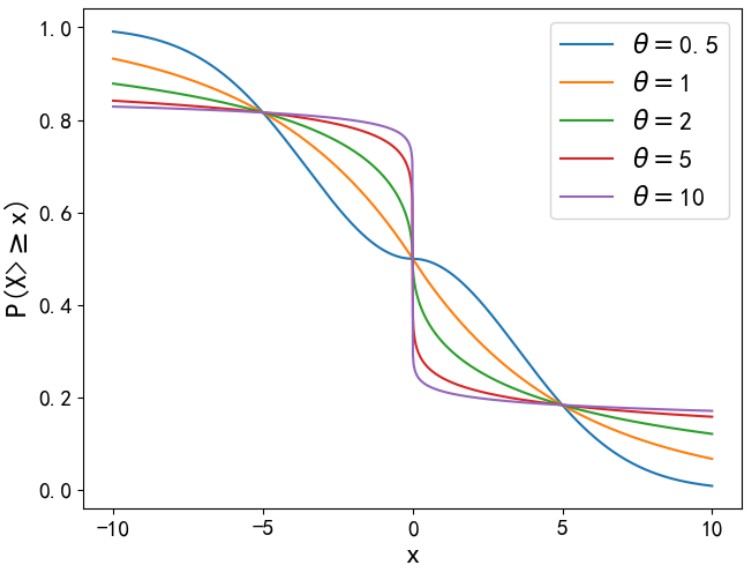

Figure 1: Some sub-Weibull survival curves with varying tail parameters $\theta$

Note that, Definition 3.6 in the main paper is just an equivalent definition and the rest equivalent definitions are listed as follows.

**Proposition B.1.** *(Vladimirova et al., 2020; Li & Liu, 2022) Let $X$ be a random variable. Then the following properties are equivalent:*

- *$\exists K_1 > 0$, the tails of $X$ satisfy $\mathbb{P}(|X \geq x) \leq 2 \exp -(x/K_1)^{1/\theta}$ for all $x \geq 0$.*

- *$\exists K_2 > 0$, the moments of $X$ satisfy $\|X\|_k \leq K_2 k^\theta$ for all $k \geq 1$.*

- *$\exists K_3 > 0$, the MGF of $|X^{1/\theta}|$ satisfies $\mathbb{E}\left[\exp\left((\lambda|X|)^{1/\theta}\right)\right] \leq \exp\left((\lambda K_3)^{1/\theta}\right)$ for all $\lambda$ such that $0 < \lambda \leq 1/K_3$,*

*where the above parameters $K_1, K_2, K_3$ and the parameter $K$ in Definition 3.6 differ each by a constant that only depends on $\theta$.*

## C    PROOFS OF MAIN RESULTS

### C.1    LEMMAS

In this subsection, we recall some technical lemmas used in our proofs.

**Lemma C.1.** *(Li & Liu, 2022). If the function $f$ is $\beta$-smooth (Assumption 3.7 (b)), then we have for any $z, \tilde{z}$,*

$$f(w; z, \tilde{z}) - f(w'; z, \tilde{z}) \leq \langle w - w', \nabla f(w'; z, \tilde{z}) \rangle + \frac{1}{2}\beta\|w - w'\|^2$$

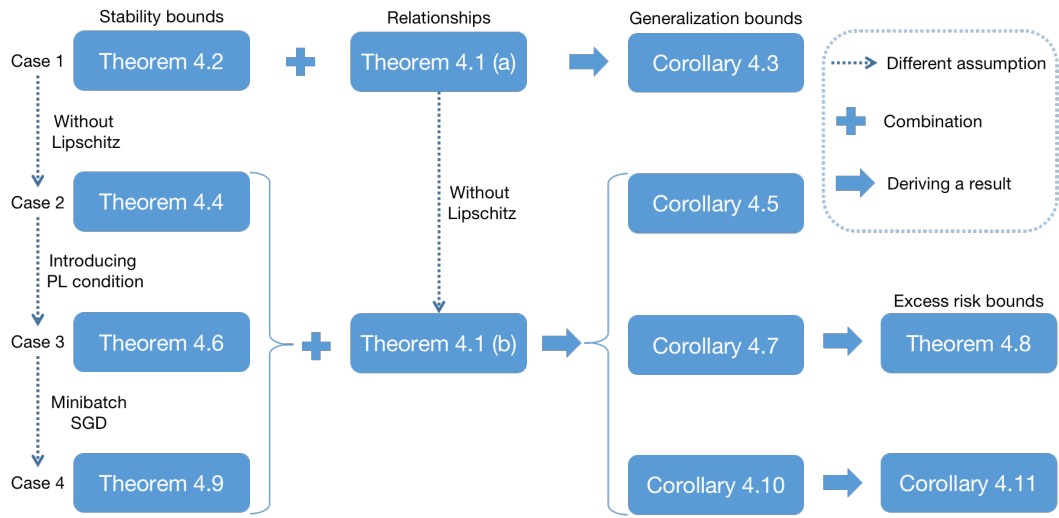

Figure 2: The proof sketch for our results in the main text

*and*

$$\frac{1}{2\beta}\|\nabla f(w; z, \tilde{z})\|^2 \leq f(w; z, \tilde{z}) - \inf_{w'} f(w'; z, \tilde{z}) \leq f(w; z, \tilde{z}).$$

**Lemma C.2.** *(Li & Liu, 2022). Let $e$ be the base of the natural logarithm. The following inequalities hold:*

*(a) if $\alpha \in (0, 1)$, then $\sum\limits_{k=1}^{t} k^{-\alpha} \leq t^{1-\alpha}/(1-\alpha)$;*

*(b) if $\alpha = 1$, then $\sum\limits_{k=1}^{t} k^{-\alpha} \leq \log(et)$;*

*(c) if $\alpha > 1$, then $\sum\limits_{k=1}^{t} k^{-\alpha} \leq \frac{\alpha}{\alpha-1}$;*

*(d) $\sum\limits_{k=1}^{t} \frac{1}{k+k_0} \leq \log(t+1)$, where $k_0 \geq 1$.*

**Lemma C.3.** *(Madden et al., 2020). Assume $X$ is $K$-sub-Weibull($\theta$), then*

*(a) $\mathbb{E}[|X|^p] \leq 2\Gamma(\theta p + 1)K^p$, where $p > 0, \Gamma(x) = \int_0^\infty t^{x-1}e^{-t}dt$. In particular, $\mathbb{E}[|X|] \leq 2\Gamma(\theta+1)K$ and $\mathbb{E}[X^2] \leq 2\Gamma(2\theta+1)K^2$;*

*(b) $\|X\|_p \leq (2\theta)^\theta K p^\theta$, where $p \geq 1/\theta$. In particular, $\|X\|_2 \leq (4\theta)^\theta K, \theta \geq 1/2$.*

**Lemma C.4.** *(Vladimirova et al., 2020; Madden et al., 2020; Li & Liu, 2022) Suppose $X_1, ..., X_n$ are sub-Weibull($\theta$) with respective parameters $K_1, ..., K_n$. Then, for $\forall t \geq 0$, with probability $1 - \delta$, we have*

$$P\left(\left|\sum_{i=1}^{n} X_i\right| \geq t\right) \leq 2\exp\left(-\left(\frac{t}{g(\theta)\sum\limits_{i=1}^{n} K_i}\right)^{1/\theta}\right),$$

## C.2 PROOF OF THEOREM 4.1

The proof of Theorem 4.1 is similar to Theorem 2 of Lei & Ying (2020) and Theorem 1 of Lei et al. (2021b). For completeness, we also provide the detailed proof here.

**Proof of Theorem 4.1**: (a) According to the symmetry, triangular inequality, $L$-Lipschitz continuity and $\ell_1$ on-average model stability, we deduce that

$$
|\mathbb{E}[F(A(S)) - F_S(A(S))]| = \left| \frac{1}{n(n-1)} \sum_{\substack{i,j \in [n], \\ i \neq j}} \mathbb{E}[F(A(S_{i,j})) - F_S(A(S))] \right|
$$

$$
= \left| \frac{1}{n(n-1)} \sum_{\substack{i,j \in [n], \\ i \neq j}} \mathbb{E}[f(A(S_{i,j}); z_i, z_j) - f(A(S); z_i, z_j)] \right|
$$

$$
\leq \frac{1}{n(n-1)} \sum_{\substack{i,j \in [n], \\ i \neq j}} \mathbb{E}[|f(A(S_{i,j}); z_i, z_j) - f(A(S); z_i, z_j)|]
$$

$$
\leq \frac{L}{n(n-1)} \sum_{\substack{i,j \in [n], \\ i \neq j}} \mathbb{E}[\|A(S_{i,j}) - A(S)\|] \leq L\epsilon.
$$

This proves Part (a).

(b) Let $g(w) = f(w) - F_S(w)$. From Lemma C.3 (b) and Assumption 3.7 (a), it is obvious that, for any $w, w' \in \mathcal{W}, w \neq w'$,

$$
\|\nabla g(w)\| \leq (4\theta)^\theta K,
$$

which means

$$
|g(w) - g(w')| \leq (4\theta)^\theta K \|w - w'\|,
$$

that is,

$$
|f(w) - F_S(w) - (f(w') - F_{S'}(w'))| \leq (4\theta)^\theta K \|w - w'\|.
$$

Then,

$$
\mathbb{E}[|f(w) - F_S(w) - (f(w') - F_{S'}(w'))|] \leq (4\theta)^\theta K \mathbb{E}[\|w - w'\|].
$$

We can derive the following relationship

$$
|\mathbb{E}[F(A(S)) - F_S(A(S))]|
$$

$$
\leq \frac{1}{n(n-1)} \sum_{\substack{i,j \in [n], \\ i \neq j}} \mathbb{E}[|f(A(S_{i,j}); z_i, z_j) - f(A(S); z_i, z_j)|]
$$

$$
\leq \frac{1}{n(n-1)} \sum_{\substack{i,j \in [n], \\ i \neq j}} \Big( \mathbb{E}[|f(A(S_{i,j}); z_i, z_j) - F_{S_{i,j}}(A(S_{i,j})) - (f(A(S); z_i, z_j) - F_S(A(S)))|]
$$

$$
+ \mathbb{E}\left[ |F_{S_{i,j}}(A(S_{i,j})) - F_S(A(S))| \right] \Big)
$$

$$
\leq \frac{1}{n(n-1)} \sum_{\substack{i,j \in [n], \\ i \neq j}} \Big( (4\theta)^\theta K \mathbb{E}[\|A(S_{i,j}) - A(S)\|] \Big) + 2\mathbb{E}[F_S(A(S))]
$$

$$
\leq (4\theta)^\theta K \epsilon + 2\mathbb{E}[F_S(A(S))].
$$

The stated result in Part (b) is proved. $\qquad \square$

C.3 PROOF OF THEOREM 4.2

**Proof of Theorem 4.2**: We know that the two data sets $S$ and $S_{i,j}$ are only different in two samples. Thus, without loss of generality, we can let $S_{i,j} = S_{n-1,n}$. In the following, we consider two cases of the sampled indexes $i_t, j_t$ at the $t$-th iteration of pairwise SGD. In the first case, the sampled samples from $S$ and $S_{n-1,n}$ are the same, i.e. $i_t \in [n-2]$ and $j_t \in [n-2], i_t \neq j_t$, and we obtain that

$$
\begin{aligned}
\|w_{t+1} - w'_{t+1}\| &= \|w_t - \eta_t \nabla f(w_t; z_{i_t}, z_{j_t}) - w'_t + \eta_t \nabla f(w'_t; z'_{i_t}, z'_{j_t})\| \\
&\leq \|w_t - w'_t\| + \eta_t \|\nabla f(w_t; z_{i_t}, z_{j_t}) - \nabla f(w'_t; z'_{i_t}, z'_{j_t})\| \\
&= \|w_t - w'_t\| + \eta_t \|\nabla f(w_t; z_{i_t}, z_{j_t}) - \nabla f(w'_t; z_{i_t}, z_{j_t})\| \\
&\leq \|w_t - w'_t\| + \eta_t \beta \|w_t - w'_t\| = (1 + \eta_t \beta)\|w_t - w'_t\|,
\end{aligned}
$$

where the last inequality uses the $\beta$-smoothness of $f$.

In the other cases, the sampled samples from $S$ and $S_{i,j}$ are different at least for one pair, i.e. $i_t, j_t \in \{n-1, n\}$ or $i_t \in [n-2], j_t \in \{n-1, n\}$ or $j_t \in [n-2], i_t \in \{n-1, n\}, i_t \neq j_t$, and we can deduce that

$$
\begin{aligned}
\|w_{t+1} - w'_{t+1}\| &= \|w_t - \eta_t \nabla f(w_t; z_{i_t}, z_{j_t}) - w'_t + \eta_t \nabla f(w'_t; z'_{i_t}, z'_{j_t})\| \\
&\leq \|w_t - w'_t\| + \eta_t \|\nabla f(w_t; z_{i_t}, z_{j_t}) - \nabla f(w'_t; z'_{i_t}, z'_{j_t})\| \\
&\leq \|w_t - w'_t\| + \eta_t \left( \|\nabla f(w_t; z_{i_t}, z_{j_t})\| + \|\nabla f(w'_t; z'_{i_t}, z'_{j_t})\| \right) \\
&\leq \|w_t - w'_t\| + 2\eta_t L,
\end{aligned}
$$

where the last inequality follows from the $L$-Lipschitz continuity of loss function. Combining the above recursive inequalities and taking expectation with respect to (w.r.t.) the indexes of the selected samples $i_t$ and $j_t$, we derive that

$$
\begin{aligned}
&\mathbb{E}_{i_t, j_t}[\|w_{t+1} - w'_{t+1}\|] \\
&\leq (1 + \eta_t \beta)\|w_t - w'_t\|\mathbb{E}_{i_t, j_t}[\mathbb{I}[i_t \in [n-2] \text{ and } j_t \in [n-2], i_t \neq j_t]] + (\|w_t - w'_t\| + 2\eta_t L) \\
&\quad \mathbb{E}_{i_t, j_t}[\mathbb{I}[i_t, j_t \in \{n-1, n\} \text{ or } i_t \in [n-2], j_t \in \{n-1, n\} \text{ or } j_t \in [n-2], i_t \in \{n-1, n\}, i_t \neq j_t]] \\
&\leq (1 + \eta_t \beta)\|w_t - w'_t\|\text{Prob}\{i_t \in [n-2] \text{ and } j_t \in [n-2], i_t \neq j_t\} + (\|w_t - w'_t\| + 2\eta_t L) \\
&\quad \text{Prob}\{i_t, j_t \in \{n-1, n\} \text{ or } i_t \in [n-2], j_t \in \{n-1, n\} \text{ or } j_t \in [n-2], i_t \in \{n-1, n\}, i_t \neq j_t\} \\
&\leq \frac{(n-2)(n-3)}{n(n-1)}(1 + \eta_t \beta)\|w_t - w'_t\| + \frac{4n-6}{n(n-1)}(\|w_t - w'_t\| + 2\eta_t L) \\
&\leq (1 + \eta_t \beta)\|w_t - w'_t\| + \frac{8n-12}{n(n-1)}\eta_t L,
\end{aligned}
$$

where the first inequality is due to the independence on $i_t$ and $j_t$ for $\|w_t - w'_t\|$. And then taking expectation w.r.t. all rest randomness to get that

$$
\mathbb{E}[\|w_{t+1} - w'_{t+1}\|] \leq (1 + \eta_t \beta)\mathbb{E}[\|w_t - w'_t\|] + \frac{8n-12}{n(n-1)}\eta_t L. \tag{7}
$$

Similarly, without Assumption 3.7 (a), it is easy to get that

$$
\mathbb{E}[\|w_{t+1} - w'_{t+1}\|] \leq (1 + \eta_t \beta)\mathbb{E}[\|w_t - w'_t\|] + \frac{8n-12}{n(n-1)}\eta_t \mathbb{E}[\|\nabla f(w_t; z_{i_t}, z_{j_t})\|], \tag{8}
$$

where $i_t, j_t \in \{n-1, n\}$ or $i_t \in [n-2], j_t \in \{n-1, n\}$ or $j_t \in [n-2], i_t \in \{n-1, n\}, i_t \neq j_t$.

For Equation (7), taking summation from $t = 1$ to $T - 1$, we can deduce that

$$
\begin{aligned}
\mathbb{E}[\|w_T - w'_T\|] &\leqslant \sum_{t=1}^{T-1} \left( \prod_{k=t+1}^{T-1} (1 + \eta_k \beta) \right) \frac{8n-12}{n(n-1)}\eta_t L \\
&\leqslant \sum_{t=1}^{T-1} \exp\left( \sum_{k=t+1}^{T-1} \eta_k \beta \right) \frac{8n-12}{n(n-1)}\eta_t L
\end{aligned}
$$

$$
\leqslant \sum_{t=1}^{T-1} \exp\left(\beta\eta_1 \sum_{k=1}^{T-1} k^{-1}\right) \frac{8n-12}{n(n-1)}\eta_t L
$$

$$
\leqslant \sum_{t=1}^{T-1} \exp\left(\beta\eta_1 \log(e(T-1))\right) \frac{8n-12}{n(n-1)}\eta_t L
$$

$$
= \sum_{t=1}^{T-1} (e(T-1))^{\beta\eta_1} \frac{8n-12}{n(n-1)}\eta_t L
$$

$$
= \frac{(8n-12)L(e(T-1))^{\beta\eta_1}\sum_{t=1}^{T-1}\eta_t}{n(n-1)}
$$

$$
\leqslant \frac{(8n-12)\eta_1 L(e(T-1))^{\beta\eta_1}}{n(n-1)} \log(e(T-1))
$$

$$
\leq \mathcal{O}\left(\frac{LT^{\frac{1}{2}}\log T}{\beta n}\right),
$$

where the second inequality uses the fact that $1 + x \leq \exp(x)$, the third inequality is caused by $\eta_t\beta \geq 0$, and the last two inequalities are due to Lemma C.2 (b). The desired result $\mathbb{E}[\|w_T - w'_T\|] \leq \mathcal{O}\left((\beta n)^{-1}LT^{\frac{1}{2}}\log T\right)$ follows since $\beta\eta_1 \leq \frac{1}{2}$. $\qquad\square$

**Proof of Corollary 4.3**: Based on Theorem 4.1 (a) and Theorem 4.2, we can get that

$$
|\mathbb{E}[F(w_T) - F_S(w_T)]| \leq \frac{L}{n(n-1)} \sum_{\substack{i,j\in[n],\\ i\neq j}} \mathbb{E}\left[\|w_T - w'_T\|\right]
$$

$$
\leq \frac{L}{n(n-1)} \sum_{\substack{i,j\in[n],\\ i\neq j}} \frac{(8n-12)}{n(n-1)}\eta_1 L(e(T-1))^{\beta\eta_1} \log(e(T-1))
$$

$$
= \frac{(8n-12)}{n(n-1)}\eta_1 L^2 (e(T-1))^{\beta\eta_1} \log(e(T-1))
$$

$$
\leq \mathcal{O}\left(\frac{L^2 T^{\frac{1}{2}}\log T}{\beta n}\right).
$$

This completes the proof. $\qquad\square$

## C.4 Proof of Theorem 4.4

**Proof of Theorem 4.4**: Without loss of generality, we can let $S_{i,j} = S_{n-1,n}$. According to Lemma C.1 (1), $(a+b)^2 \leq 2(a^2 + b^2), \forall a, b \in \mathbb{R}$ and $\beta\eta_t^2 - \eta_t \leq -\frac{\eta_t}{2}$ with the assumption $\eta_t \leq \frac{1}{2\beta}$, we have

$$
F_S(w_{t+1}) - F_S(w_t)
$$

$$
\leqslant \langle w_{t+1} - w_t, \nabla F_S(w_t)\rangle + \frac{1}{2}\beta\|w_{t+1} - w_t\|^2
$$

$$
= -\eta_t \langle \nabla f(w_t; z_{i_t}, z_{j_t}) - \nabla F_S(w_t), \nabla F_S(w_t)\rangle - \eta_t \|\nabla F_S(w_t)\|^2 + \frac{1}{2}\beta\eta_t^2 \|\nabla f(w_t; z_{i_t}, z_{j_t})\|^2
$$

$$
\leqslant -\eta_t \langle \nabla f(w_t; z_{i_t}, z_{j_t}) - \nabla F_S(w_t), \nabla F_S(w_t)\rangle - \left(\eta_t - \beta\eta_t^2\right)\|\nabla F_S(w_t)\|^2
$$
$$
+ \beta\eta_t^2 \|\nabla f(w_t; z_{i_t}, z_{j_t}) - \nabla F_S(w_t)\|^2
$$

$$
\leqslant -\eta_t \langle \nabla f(w_t; z_{i_t}, z_{j_t}) - \nabla F_S(w_t), \nabla F_S(w_t)\rangle - \frac{1}{2}\eta_t \|\nabla F_S(w_t)\|^2
$$
$$
+ \beta\eta_t^2 \|\nabla f(w_t; z_{i_t}, z_{j_t}) - \nabla F_S(w_t)\|^2.
$$

Then, by a summation from $t' = 1$ to $t$ and triangular inequality, we get that

$$
F_S(w_{t+1})
$$

$$\leq F_S(w_1) - \frac{1}{2}\sum_{t'=1}^{t}\eta_{t'}\|\nabla F_S(w_{t'})\|^2 - \sum_{t'=1}^{t}\eta_{t'}\left\langle\nabla f(w_{t'};z_{i_{t'}},z_{j_{t'}}) - \nabla F_S(w_{t'}),\nabla F_S(w_{t'})\right\rangle$$

$$+ \sum_{t'=1}^{t}\beta\eta_{t'}^2\|\nabla f\left(w_{t'};z_{i_{t'}},z_{j_{t'}}\right) - \nabla F_S\left(w_{t'}\right)\|^2$$

$$\leq F_S(w_1) - \frac{1}{2}\sum_{t'=1}^{t}\eta_{t'}\|\nabla F_S(w_{t'})\|^2 + \sum_{t'=1}^{t}\beta\eta_{t'}^2\|\nabla f\left(w_{t'};z_{i_{t'}},z_{j_{t'}}\right) - \nabla F_S\left(w_{t'}\right)\|^2$$

$$+ \sum_{t'=1}^{t}\eta_{t'}\left(\frac{1}{2}\|\nabla f(w_{t'};z_{i_{t'}},z_{j_{t'}}) - \nabla F_S(w_{t'})\|^2 + \frac{1}{2}\|\nabla F_S(w_{t'})\|^2\right)$$

$$= F_S(w_1) + \sum_{t'=1}^{t}\left(\frac{1}{2}\eta_{t'} + \beta\eta_{t'}^2\right)\|\nabla f\left(w_{t'};z_{i_{t'}},z_{j_{t'}}\right) - \nabla F_S\left(w_{t'}\right)\|^2. \tag{9}$$

We consider the two parts of the last term $\sum_{t'=1}^{t}\left(\frac{1}{2}\eta_{t'} + \beta\eta_{t'}^2\right)\|\nabla f\left(w_{t'};z_{i_{t'}},z_{j_{t'}}\right) - \nabla F_S\left(w_{t'}\right)\|^2$, respectively. Firstly, for $\frac{1}{2}\eta_{t'}\|\nabla f\left(w_{t'};z_{i_{t'}},z_{j_{t'}}\right) - \nabla F_S\left(w_{t'}\right)\|^2$, since $\nabla f\left(w_{t'};z_{i_{t'}},z_{j_{t'}}\right) - \nabla F_S\left(w_{t'}\right)$ is a sub-Weibull random variable, i.e., $\nabla f\left(w_{t'};z_{i_{t'}},z_{j_{t'}}\right) - \nabla F_S\left(w_{t'}\right) \sim subW(\theta,K)$, we get

$$\mathbb{E}\left[\exp\left(\frac{\frac{1}{2}\eta_{t'}\|\nabla f\left(w_{t'};z_{i_{t'}},z_{j_{t'}}\right) - \nabla F_S\left(w_{t'}\right)\|^2}{\frac{1}{2}\eta_{t'}K^2}\right)^{\frac{1}{2\theta}}\right] \leq 2,$$

which means that $\frac{1}{2}\eta_t(\nabla f\left(w_t;z_{i_t},z_{j_t}\right) - \nabla F_S\left(w_t\right))^2 \sim subW\left(2\theta,\frac{1}{2}\eta_t K^2\right)$. Similarly, $\beta\eta_t^2(\nabla f\left(w_t;z_{i_t},z_{j_t}\right) - \nabla F_S\left(w_t\right))^2 \sim subW(2\theta,\beta\eta_t^2 K^2)$. Based on Lemma C.2 (b), (c), Lemma C.3 (a) and sub-Weibull noise, we take expectation w.r.t. all randomness to obtain that

$$\mathbb{E}[F_S(w_{t+1})] \leq \mathbb{E}\left[F_S(w_1) + \sum_{t'=1}^{t}\left(\frac{1}{2}\eta_{t'} + \beta\eta_{t'}^2\right)\|\nabla f\left(w_{t'};z_{i_{t'}},z_{j_{t'}}\right) - \nabla F_S\left(w_{t'}\right)\|^2\right]$$

$$\leq \mathbb{E}[F_S(w_1)] + \sum_{t'=1}^{t}\left(\Gamma(2\theta+1)\eta_{t'}K^2 + 2\beta\Gamma(2\theta+1)\eta_{t'}^2 K^2\right)$$

$$\leq F(w_1) + (2\beta)^{-1}\Gamma(2\theta+1)K^2\left(\log(et) + 1\right).$$

According to Lemma C.1 (2), the following inequality holds

$$\mathbb{E}[\|\nabla f(w_t;z_{i_t},z_{j_t})\|^2] \leq 2\beta\mathbb{E}[F_S(w_t)] \leq 2\beta F(w_1) + \Gamma(2\theta+1)K^2(\log(e(t-1)) + 1).$$

To simplify the subsequent proof process, we let $\tau(t) = 2\beta\mathbb{E}[F_S(w_t)] \leq 2\beta F(w_1) + \Gamma(2\theta+1)K^2(\log(e(t-1)) + 1)$. From Equation (8), we know that

$$\mathbb{E}[\|w_{t+1} - w_{t+1}'\|] \leq (1 + \eta_t\beta)\mathbb{E}[\|w_t - w_t'\|] + \frac{8n-12}{n(n-1)}\eta_t\mathbb{E}[\|\nabla f(w_t;z_{i_t},z_{j_t})\|],$$

where $i_t, j_t \in \{n-1,n\}$ or $i_t \in [n-2], j_t \in \{n-1,n\}$ or $j_t \in [n-2], i_t \in \{n-1,n\}, i_t \neq j_t$. Then,

$$\mathbb{E}[\|w_{t+1} - w_{t+1}'\|] \leq (1 + \eta_t\beta)\mathbb{E}[\|w_t - w_t'\|] + \frac{8n-12}{n(n-1)}\eta_t\sqrt{\tau(t)}.$$

Similar to the proof of Theorem 4.2, taking summation from $t = 1$ to $T - 1$, we have

$$\mathbb{E}[\|w_T - w_T'\|]$$

$$\leq \sum_{t=1}^{T-1}\left(\prod_{k=t+1}^{T-1}(1 + \eta_k\beta)\right)\frac{8n-12}{n(n-1)}\eta_t\sqrt{\tau(t)}$$

$$\leq \sum_{t=1}^{T-1}\left(\prod_{k=t+1}^{T-1}(1 + \eta_k\beta)\right)\frac{8n-12}{n(n-1)}\eta_t\sqrt{\tau(T-1)}$$

$$\leq \sum_{t=1}^{T-1} \exp\left(\sum_{k=t+1}^{T-1} \eta_k \beta\right) \frac{8n-12}{n(n-1)} \eta_t \sqrt{\tau(T-1)}$$

$$\leq \sum_{t=1}^{T-1} \exp\left(\sum_{k=1}^{T-1} \eta_k \beta\right) \frac{8n-12}{n(n-1)} \eta_t \sqrt{\tau(T-1)}$$

$$\leq \sum_{t=1}^{T-1} \exp\left(\eta_1 \beta \log(e(T-1))\right) \frac{8n-12}{n(n-1)} \eta_t \sqrt{\tau(T-1)}$$

$$\leq \frac{8n-12}{n(n-1)} (e(T-1))^{\beta\eta_1} \sqrt{\tau(T-1)} \sum_{t=1}^{T-1} \eta_t$$

$$\leq \frac{8n-12}{n(n-1)} (e(T-1))^{\beta\eta_1} \eta_1 \sqrt{\tau(T-1)} \log(e(T-1))$$

$$= \frac{8n-12}{n(n-1)} \eta_1 (e(T-1))^{\beta\eta_1} \sqrt{2\beta F(w_1) + \Gamma(2\theta+1)K^2(\log(e(T-1))+1)} \log(e(T-1))$$

$$\leq \mathcal{O}\left(\frac{\sqrt{\Gamma(2\theta+1)} T^{\frac{1}{2}}(\log T)^{\frac{3}{2}}}{n\beta}\right).$$

$\square$

**Proof of Corollary 4.5**: Combining Theorem 4.1 (b) and Theorem 4.4, we have

$$|\mathbb{E}[F(w_T) - F_S(w_T)]|$$

$$\leq \frac{1}{n(n-1)} \sum_{\substack{i,j\in[n],\\i\neq j}} \left((4\theta)^\theta K\mathbb{E}[\|w_T - w_T'\|]\right) + 2\mathbb{E}[F_S(w_T))]$$

$$\leq \frac{8n-12}{n(n-1)} (4\theta)^\theta K\eta_1 (e(T-1))^{\beta\eta_1} \sqrt{2\beta F(w_1) + \Gamma(2\theta+1)K^2(\log(e(T-1))+1)}$$
$$\log(e(T-1)) + 2\mathbb{E}[F_S(w_T)]$$

$$\leq \mathcal{O}\left(\frac{(4\theta)^\theta \sqrt{\Gamma(2\theta+1)} T^{\frac{1}{2}}(\log T)^{\frac{3}{2}}}{n\beta} + \mathbb{E}[F_S(w_T)]\right).$$

When $\mathbb{E}[F_S(w_T)] = \mathcal{O}\left(n^{-1}\right)$, we derive

$$|\mathbb{E}[F(w_T) - F_S(w_T)]| \leq \mathcal{O}\left(\frac{(4\theta)^\theta \sqrt{\Gamma(2\theta+1)} T^{\frac{1}{2}}(\log T)^{\frac{3}{2}}}{n\beta}\right).$$

The proof is complete. $\square$

### C.5 PROOF OF THEOREMS 4.6 AND 4.8

**Proof of Theorem 4.6**: Without loss of generality, we can let $S_{i,j} = S_{n-1,n}$. According to Lemma C.1 (1), we have

$$F_S(w_{t+1}) - F_S(w_t)$$

$$\leq \langle w_{t+1} - w_t, \nabla F_S(w_t)\rangle + \frac{1}{2}\beta\|w_{t+1} - w_t\|^2$$

$$= -\eta_t \langle \nabla f(w_t; z_{i_t}, z_{j_t}) - \nabla F_S(w_t), \nabla F_S(w_t)\rangle - \eta_t\|\nabla F_S(w_t)\|^2 + \frac{1}{2}\beta\eta_t^2\|\nabla f(w_t; z_{i_t}, z_{j_t})\|^2$$

$$\leq -\eta_t\langle \nabla f(w_t; z_{i_t}, z_{j_t}) - \nabla F_S(w_t), \nabla F_S(w_t)\rangle - (\eta_t - \beta\eta_t^2)\|\nabla F_S(w_t)\|^2$$
$$+ \beta\eta_t^2\|\nabla f(w_t; z_{i_t}, z_{j_t}) - \nabla F_S(w_t)\|^2$$

$$\leq \eta_t\|\nabla f(w_t; z_{i_t}, z_{j_t}) - \nabla F_S(w_t)\|\|\nabla F_S(w_t)\| - (\eta_t - \beta\eta_t^2)\|\nabla F_S(w_t)\|^2$$
$$+ \beta\eta_t^2\|\nabla f(w_t; z_{i_t}, z_{j_t}) - \nabla F_S(w_t)\|^2$$

$$\leq \frac{1}{2}\eta_t \|\nabla f(w_t; z_{i_t}, z_{j_t}) - \nabla F_S(w_t)\|^2 + \frac{1}{2}\eta_t \|\nabla F_S(w_t)\|^2 - (\eta_t - \beta\eta_t^2)\|\nabla F_S(w_t)\|^2$$
$$+ \beta\eta_t^2 \|\nabla f(w_t; z_{i_t}, z_{j_t}) - \nabla F_S(w_t)\|^2$$
$$= \left(\beta\eta_t^2 - \frac{1}{2}\eta_t\right)\|\nabla F_S(w_t)\|^2 + \left(\beta\eta_t^2 + \frac{1}{2}\eta_t\right)\|\nabla f(w_t; z_{i_t}, z_{j_t}) - \nabla F_S(w_t)\|^2$$
$$\leq -\frac{1}{4}\eta_t \|\nabla F_S(w_t)\|^2 + \left(\beta\eta_t^2 + \frac{1}{2}\eta_t\right)\|\nabla f(w_t; z_{i_t}, z_{j_t}) - \nabla F_S(w_t)\|^2$$
$$\leq -\frac{1}{2}\eta_t\mu\left(F_S(w_t) - F_S(w(S))\right) + \left(\beta\eta_t^2 + \frac{1}{2}\eta_t\right)\|\nabla f(w_t; z_{i_t}, z_{j_t}) - \nabla F_S(w_t)\|^2,$$

where the fourth inequality follows from that $ab \leq \frac{1}{2}a^2 + \frac{1}{2}b^2, \forall a, b > 0$, the fifth inequality is caused by the fact that $\eta_t \leq \frac{1}{4\beta}$, and the last inequality is due to Assumption 3.9. Taking expectation w.r.t. all randomness, based on Lemma C.3 (a), we obtain that

$$\mathbb{E}[F_S(w_{t+1}) - F_S(w_t)]$$
$$\leq -\frac{1}{2}\mu\eta_t\mathbb{E}\left[F_S(w_t) - F_S(w(S))\right] + \left(\beta\eta_t^2 + \frac{1}{2}\eta_t\right)\mathbb{E}\left[\|\nabla f(w_t; z_{i_t}, z_{j_t}) - \nabla F_S(w_t)\|^2\right]$$
$$\leq -\frac{1}{2}\mu\eta_t\mathbb{E}\left[F_S(w_t) - F_S(w(S))\right] + (2\beta\eta_t^2 + \eta_t)\Gamma(2\theta + 1)K^2,$$

which is similar to the proof of Theorem 4.4. Then,

$$\mathbb{E}[F_S(w_{t+1}) - F_S(w(S))]$$
$$\leq \left(1 - \frac{1}{2}\mu\eta_t\right)\mathbb{E}[F_S(w_t) - F_S(w(S))] + (2\beta\eta_t^2 + \eta_t)\Gamma(2\theta + 1)K^2$$
$$\leq \prod_{i=1}^{t}\left(1 - \frac{1}{2}\mu\eta_i\right)\mathbb{E}[F_S(w_1) - F_S(w(S))] + \sum_{i=1}^{t}\left(2\beta\eta_i^2 + \eta_i\right)\Gamma(2\theta + 1)K^2$$
$$\leq \prod_{i=1}^{t}\left(1 - \frac{1}{2}\mu\eta_i\right)\mathbb{E}[F_S(w_1) - F_S(w(S))] + \Gamma(2\theta + 1)K^2\sum_{i=1}^{t}(2\beta\eta_i^2 + \eta_i)$$
$$= \prod_{i=1}^{t}\left(1 - \frac{1}{2}\mu\eta_i\right)\mathbb{E}[F_S(w_1) - F_S(w(S))] + 2\beta\Gamma(2\theta + 1)K^2\eta_1^2\sum_{i=1}^{t}\frac{1}{i^2} + \Gamma(2\theta + 1)K^2\eta_1\sum_{i=1}^{t}\frac{1}{i}$$
$$\leq \prod_{i=1}^{t}\left(1 - \frac{1}{2}\mu\eta_i\right)\mathbb{E}[F_S(w_1) - F_S(w(S))] + 2\beta\Gamma(2\theta + 1)K^2\eta_1^2 + \Gamma(2\theta + 1)K^2\eta_1\log(et),$$

where the last inequality is due to Lemma C.2 (b), (c). For convenience, we let $a_1 = 1 - \prod_{i=1}^{t}\left(1 - \frac{1}{2}\mu\eta_i\right)$. Then, it follows from Lemma C.1 (2) that

$$\mathbb{E}[\|\nabla f(w_t; z_{i_t}, z_{j_t})\|^2]$$
$$\leq 2\beta\mathbb{E}[F_S(w_t)]$$
$$= 2\beta\left((1 - a_1)\mathbb{E}[F_S(w_1)] + a_1\mathbb{E}[F_S(w(S))] + 4\beta\Gamma(2\theta + 1)K^2\eta_1^2 + \Gamma(2\theta + 1)K^2\eta_1\log(et)\right).$$

To simplify the subsequent proof process, we let

$$\tilde{\tau}(t) = 2\beta\left((1 - a_1)\mathbb{E}[F_S(w_1)] + a_1\mathbb{E}[F_S(w(S))] + 4\beta\Gamma(2\theta + 1)K^2\eta_1^2 + \Gamma(2\theta + 1)K^2\eta_1\log(et)\right).$$

From Equation (8), we know that

$$\mathbb{E}[\|w_{t+1} - w'_{t+1}\|] \leq (1 + \eta_t\beta)\mathbb{E}[\|w_t - w'_t\|] + \frac{8n - 12}{n(n-1)}\eta_t\mathbb{E}\left[\|\nabla f(w_t; z_{i_t}, z_{j_t})\|\right],$$

where $i_t, j_t \in \{n-1, n\}$ or $i_t \in [n-2], j_t \in \{n-1, n\}$ or $j_t \in [n-2], i_t \in \{n-1, n\}, i_t \neq j_t$. Then, taking summation from $t = 1$ to $T - 1$, we get that

$$\mathbb{E}[\|w_T - w'_T\|]$$

$$\leq \sum_{t=1}^{T-1} \left( \prod_{k=t+1}^{T-1} (1 + \eta_k \beta) \right) \frac{8n - 12}{n(n-1)} \eta_t \sqrt{\tilde{\tau}(t)}$$

$$\leq \sum_{t=1}^{T-1} \exp \left( \beta \eta_1 \sum_{k=t+1}^{T-1} k^{-1} \right) \frac{8n - 12}{n(n-1)} \eta_t \sqrt{\tilde{\tau}(T-1)}$$

$$\leq \sum_{t=1}^{T-1} \exp \left( \beta \eta_1 \sum_{k=1}^{T-1} k^{-1} \right) \frac{8n - 12}{n(n-1)} \eta_t \sqrt{\tilde{\tau}(T-1)}$$

$$\leq \frac{(8n - 12)\eta_1}{n(n-1)} (e(T-1))^{\beta \eta_1} \sqrt{\tilde{\tau}(T-1)} \sum_{t=1}^{T-1} t^{-1}$$

$$\leq \frac{(8n - 12)\eta_1}{n(n-1)} (e(T-1))^{\beta \eta_1} \sqrt{\tilde{\tau}(T-1)} \log(e(T-1)),$$

where the first and second inequalities are similar to Theorem 4.4, and the fourth and fifth inequalities are derived from Lemma C.2 (c), that is,

$$\mathbb{E}[\|w_T - w_T'\|]$$

$$\leq \frac{(8\sqrt{2}n - 12\sqrt{2})\sqrt{\beta}\eta_1}{n(n-1)} (e(T-1))^{\beta \eta_1} \log(e(T-1))$$

$$\sqrt{(1 - a_1)\mathbb{E}[F_S(w_1)] + a_1 \mathbb{E}[F_S(w(S))] + 4\beta \Gamma(2\theta + 1) K^2 \eta_1^2 + \Gamma(2\theta + 1) K^2 \eta_1 \log(e(T-1))}$$

$$\leq \mathcal{O} \left( \frac{1}{n} \beta^{-\frac{1}{2}} T^{\frac{1}{4}} \log T \sqrt{a_1 \mathbb{E}[F_S(w(S))] + \beta^{-1} \Gamma(2\theta + 1) \log T} \right).$$

When $\mathbb{E}[F_S(w(S))] \leq \mathcal{O}(n^{-1})$,

$$\mathbb{E}[\|w_T - w_T'\|] \leq \mathcal{O} \left( (\beta n)^{-1} T^{\frac{1}{4}} (\Gamma(2\theta + 1))^{\frac{1}{2}} (\log T)^{\frac{3}{2}} \right).$$

$\square$

**Proof of Corollary 4.7**: By Theorem 4.1 (b) and Theorem 4.6, we can get that

$$|\mathbb{E}[F(w_T) - F_S(w_T)]|$$

$$\leq 2\mathbb{E}[F_S(w_T)] + \frac{(4\theta)^\theta K}{n(n-1)} \sum_{\substack{i,j \in [n], \\ i \neq j}} \mathbb{E}[\|w_T - w_T'\|]$$

$$\leq \mathcal{O} \left( \mathbb{E}[F_S(w_T)] + \frac{1}{n} \beta^{-\frac{1}{2}} T^{\frac{1}{4}} (4\theta)^\theta \log T \sqrt{a_1 \mathbb{E}[F_S(w(S))] + \beta^{-1} \Gamma(2\theta + 1) \log T} \right).$$

When $\mathbb{E}[F_S(w(S))] \leq \mathbb{E}[F_S(w_T)] = \mathcal{O}(n^{-1})$,

$$|\mathbb{E}[F(w_T) - F_S(w_T)]| \leq \mathcal{O} \left( (\beta n)^{-1} T^{\frac{1}{4}} (4\theta)^\theta (\Gamma(2\theta + 1))^{\frac{1}{2}} (\log T)^{\frac{3}{2}} \right).$$

$\square$

**Proof of Theorem 4.8**: Without loss of generality, we can let $S_{i,j} = S_{n-1,n}$ and $J_t = \{i_t, j_t\}$. According to Lemma C.1 (1) and taking expectation w.r.t. $J_t$, we have

$$\mathbb{E}_{J_t}[F_S(w_{t+1}) - F_S(w_t)]$$

$$\leq \mathbb{E}_{J_t} \left[ \langle w_{t+1} - w_t, \partial F_S(w_t) \rangle + \frac{1}{2} \beta \|w_{t+1} - w_t\|^2 \right]$$

$$= \mathbb{E}_{J_t} \left[ -\eta_t \langle \partial f(w_t; z_{i_t}, z_{j_t}), \partial F_S(w_t) \rangle + \frac{1}{2} \beta \eta_t^2 \|\partial f(w_t; z_{i_t}, z_{j_t})\|^2 \right]$$

$$\leq -\eta_t \|\partial F_S(w_t)\|^2 + \beta \eta_t^2 \mathbb{E}_{J_t} \left[ \|\partial f(w_t; z_{i_t}, z_{j_t}) - \partial F_S(w_t)\|^2 \right] + \beta \eta_t^2 \|\partial F_S(w_t)\|^2$$

$$\leq -\frac{1}{2} \eta_t \|\partial F_S(w_t)\|^2 + \beta \eta_t^2 \mathbb{E}_{J_t} \left[ \|\partial f(w_t; z_{i_t}, z_{j_t}) - \partial F_S(w_t)\|^2 \right]$$

$$\leq - \mu\eta_t \left(F_S(w_t) - F_S(w(S))\right) + \beta\eta_t^2 \mathbb{E}_{J_t} \left[\|\partial f(w_t; z_{i_t}, z_{j_t}) - \partial F_S(w_t)\|^2\right],$$

where the second inequality is from $\mathbb{E}_{J_t} \left[-\eta_t \langle \partial f(w_t; z_{i_t}, z_{j_t}), \partial F_S(w_t)\rangle\right] = -\eta_t \|\partial F_S(w_t)\|^2$ and $(a+b)^2 \leq 2(a^2+b^2)$, the third inequality is caused by the fact that $\eta_t \leq \frac{1}{4\beta} \leq \frac{1}{2\beta}$, and the last inequality is due to Assumption 3.9. Based on Lemma C.3 (a), we obtain that

$$\mathbb{E}[F_S(w_{t+1}) - F_S(w_t)]$$
$$\leq - \mu\eta_t \mathbb{E}\left[F_S(w_t) - F_S(w(S))\right] + \beta\eta_t^2 \mathbb{E}\left[\|\nabla f(w_t; z_{i_t}, z_{j_t}) - \nabla F_S(w_t)\|^2\right]$$
$$\leq - \mu\eta_t \mathbb{E}\left[F_S(w_t) - F_S(w(S))\right] + 2\beta\eta_t^2 \Gamma(2\theta+1)K^2,$$

which is similar to the proof of Theorem 4.4. Then,

$$\mathbb{E}[F_S(w_{t+1}) - F_S(w(S))]$$
$$\leq (1 - \mu\eta_t)\mathbb{E}[F_S(w_t) - F_S(w(S))] + 2\beta\eta_t^2 \Gamma(2\theta+1)K^2$$
$$= \left(1 - \frac{\mu\eta_1}{t}\right)\mathbb{E}[F_S(w_t) - F_S(w(S))] + 2\beta\eta_1^2 \Gamma(2\theta+1)K^2 t^{-2}.$$

We multiply both sides of the above inequality by $t\left(t - \frac{1}{2}\mu\eta_1\right)$ and get

$$t\left(t - \frac{1}{2}\mu\eta_1\right)\mathbb{E}[F_S(w_{t+1}) - F_S(w(S))]$$
$$\leq \left(t - \frac{1}{2}\mu\eta_1\right)(t - \mu\eta_1)\mathbb{E}[F_S(w_t) - F_S(w(S))] + 2\beta\eta_1^2 \Gamma(2\theta+1)K^2.$$

Then, we can take a summation from $t = 1$ to $T - 1$ to get

$$(T-1)\left(T - 1 - \frac{1}{2}\mu\eta_1\right)\mathbb{E}[F_S(w_T) - F_S(w(S))]$$
$$\leq \left(1 - \frac{1}{2}\mu\eta_1\right)(1 - \mu\eta_1)\mathbb{E}[F_S(w_1) - F_S(w(S))] + 2\beta\eta_1^2 \Gamma(2\theta+1)K^2(T-1)$$
$$\leq \left(1 - \frac{1}{2}\mu\eta_1\right)(1 - \mu\eta_1)\mathbb{E}[F_S(w_1) - F_S(w(S))] + \frac{1}{8\beta}\Gamma(2\theta+1)K^2(T-1).$$

Therefore,

$$\mathbb{E}[F_S(w_T) - F_S(w(S))]$$
$$\leq \frac{(1 - \frac{1}{2}\mu\eta_1)(1 - \mu\eta_1)}{(T-1)(T - 1 - \frac{1}{2}\mu\eta_1)}\mathbb{E}[F_S(w_1) - F_S(w(S))] + \frac{\Gamma(2\theta+1)K^2}{8\beta(T - 1 - \frac{1}{2}\mu\eta_1)} = \mathcal{O}\left(\frac{\Gamma(2\theta+1)}{\beta T}\right).$$

We finish the proof of optimization error bound. Now, we prove the excess risk bound. By Corollary 4.7, we have

$$|\mathbb{E}[F(w_T) - F_S(w_T)]|$$
$$\leq \mathcal{O}\left(\mathbb{E}[F_S(w_T)] + \frac{1}{n}\beta^{-\frac{1}{2}}T^{\frac{1}{4}}(4\theta)^\theta \log T \sqrt{a_1 \mathbb{E}[F_S(w(S))] + \beta^{-1}\Gamma(2\theta+1)\log T}\right).$$

Let $\mathbb{E}[F_S(w(S))] \leq \mathbb{E}[F_S(w_T)] = \mathcal{O}\left(n^{-1}\right)$, then,

$$|\mathbb{E}[F(w_T) - F(w^*)]|$$
$$\leq |\mathbb{E}[F_S(w_T) - F_S(w(S))]| + |\mathbb{E}[F(w_T) - F_S(w_T)]|$$
$$\leq \mathcal{O}\left(\frac{\Gamma(2\theta+1)}{\beta T} + \mathbb{E}[F_S(w_T)] + \frac{1}{n}\beta^{-\frac{1}{2}}T^{\frac{1}{4}}(4\theta)^\theta \log T \sqrt{a_1 \mathbb{E}[F_S(w(S))] + \beta^{-1}\Gamma(2\theta+1)\log T}\right)$$
$$\leq \mathcal{O}\left(\frac{\Gamma(2\theta+1)}{\beta T} + (\beta n)^{-1}T^{\frac{1}{4}}(4\theta)^\theta (\Gamma(2\theta+1))^{\frac{1}{2}}(\log T)^{\frac{3}{2}}\right).$$

The proof is complete. $\qquad\square$

### C.6 PROOF OF THEOREMS 4.9 AND 4.11

**Proof of Theorem 4.9**: For general SGD, it is necessary to consider whether the disturbed sample will be drawn to update the model parameter at each iteration. As for minibatch SGD, we draw b samples per iteration and the disturbed sample can be drawn repetitively. Thus, the new formula of minibatch SGD is constructed to overcome the barrier of analyzing traditional formulas using our proof framework.

Without loss of generality, we can let $S_{i,j} = S_{n-1,n}$. Define $\alpha_{t,k,k'} = |\{m : i_{t,m} = k, j_{t,m} = k'\}|, \forall t \in \mathbb{N}, k, k' \in [n], k \neq k', m \in [b]$, where $|\cdot|$ denotes the cardinality of a set. That is $\alpha_{t,k,k'}$ is the number of indices $i$ and $j$ equal to $k$ and $k'$ in the $t$-th iteration. Then, the minibatch pairwise SGD update (6) can be reformulated as

$$w_{t+1} = w_t - \frac{\eta_t}{b} \sum_{\substack{k,k' \in [n], \\ k \neq k'}} \alpha_{t,k,k'} \nabla f(w_t; z_k, z_{k'}).$$

According to Lemma C.1 (1), we have

$$F_S(w_{t+1}) - F_S(w_t)$$

$$\leq \langle w_{t+1} - w_t, \nabla F_S(w_t)\rangle + \frac{1}{2}\beta\|w_{t+1} - w_t\|^2$$

$$\leq -\frac{\eta_t}{b} \sum_{\substack{k,k' \in [n], \\ k \neq k'}} \alpha_{t,k,k'} \langle \nabla f(w_t; z_k, z_{k'}), \nabla F_S(w_t)\rangle + \frac{\beta\eta_t^2}{2}\left\|\frac{1}{b}\sum_{\substack{k,k' \in [n], \\ k \neq k'}} \alpha_{t,k,k'} \nabla f(w_t; z_k, z_{k'})\right\|^2$$

$$= -\frac{\eta_t}{b} \sum_{\substack{k,k' \in [n], \\ k \neq k'}} \alpha_{t,k,k'} \langle \nabla f(w_t; z_k, z_{k'}) - \nabla F_S(w_t), \nabla F_S(w_t)\rangle - \frac{\eta_t}{b} \sum_{\substack{k,k' \in [n], \\ k \neq k'}} \alpha_{t,k,k'}\|\nabla F_S(w_t)\|^2$$

$$+ \frac{\beta\eta_t^2}{2}\left\|\frac{1}{b}\sum_{\substack{k,k' \in [n], \\ k \neq k'}} \alpha_{t,k,k'} \nabla f(w_t; z_k, z_{k'})\right\|^2$$

$$\leq \frac{\eta_t}{2b} \sum_{\substack{k,k' \in [n], \\ k \neq k'}} \alpha_{t,k,k'}\|\nabla f(w_t; z_k, z_{k'}) - \nabla F_S(w_t)\|^2 + \frac{\eta_t}{2b} \sum_{\substack{k,k' \in [n], \\ k \neq k'}} \alpha_{t,k,k'}\|\nabla F_S(w_t)\|^2$$

$$- \frac{\eta_t}{b} \sum_{\substack{k,k' \in [n], \\ k \neq k'}} \alpha_{t,k,k'}\|\nabla F_S(w_t)\|^2 + \frac{\beta\eta_t^2}{2}\left\|\frac{1}{b}\sum_{\substack{k,k' \in [n], \\ k \neq k'}} \alpha_{t,k,k'} \nabla f(w_t; z_k, z_{k'})\right\|^2.$$

Based on the definition of $\alpha_{t,k,k'}$, we know that $\alpha_{t,k,k'}$ is a random variable following from the binomial distribution $B\left(b, \frac{1}{n(n-1)}\right)$ with parameters $b$ and $\frac{1}{n(n-1)}$, thus, it is easy to know that $\mathbb{E}\left[\alpha_{t,k,k'}\right] = \frac{b}{n(n-1)}$ and

$$\mathbb{E}[\alpha_{t,k,k'}^2] = (\mathbb{E}\left[\alpha_{t,k,k'}\right])^2 + \text{Var}\left(\alpha_{t,k,k'}\right) = \frac{b}{n(n-1)}\left(1 + \frac{b-1}{n(n-1)}\right) \geq (\mathbb{E}\left[\alpha_{t,k,k'}\right])^2.$$

For simplicity, we define $J_t = \{(i_{t,1}, j_{t,1}), (i_{t,2}, j_{t,2}), ..., (i_{t,b}, j_{t,b})\}, t \in \mathbb{N}$. Taking conditional expectation w.r.t. $J_t$, we get that

$$\mathbb{E}_{J_t}[F_S(w_{t+1}) - F_S(w_t)]$$

$$\leq -\frac{\eta_t}{2b} \sum_{\substack{k,k' \in [n], \\ k \neq k'}} \mathbb{E}_{J_t}[\alpha_{t,k,k'}]\|\nabla F_S(w_t)\|^2 + \frac{\beta\eta_t^2}{2b^2}\mathbb{E}_{J_t}\left[\left\|\sum_{\substack{k,k' \in [n], \\ k \neq k'}} \alpha_{t,k,k'} \nabla f(w_t; z_k, z_{k'})\right\|^2\right]$$

$$+ \frac{\eta_t}{2b} \sum_{\substack{k,k' \in [n], \\ k \neq k'}} \mathbb{E}_{J_t}[\alpha_{t,k,k'}] \|\nabla f(w_t; z_k, z_{k'}) - \nabla F_S(w_t)\|^2$$

$$\leq - \frac{\eta_t}{2} \|\nabla F_S(w_t)\|^2 + \frac{\beta \eta_t^2}{2b^2} \mathbb{E}_{J_t} \left[\alpha_{t,1,2}^2\right] \left\| \sum_{\substack{k,k' \in [n], \\ k \neq k'}} \nabla f(w_t; z_k, z_{k'}) \right\|^2$$

$$+ \frac{\eta_t}{2n(n-1)} \sum_{\substack{k,k' \in [n], \\ k \neq k'}} \|\nabla f(w_t; z_k, z_{k'}) - \nabla F_S(w_t)\|^2$$

$$= - \frac{\eta_t}{2} \|\nabla F_S(w_t)\|^2 + \frac{\beta \eta_t^2}{2b^2} \frac{b}{n(n-1)} \left(1 + \frac{b-1}{n(n-1)}\right) \left\| \sum_{\substack{k,k' \in [n], \\ k \neq k'}} \nabla f(w_t; z_k, z_{k'}) \right\|^2$$

$$+ \frac{\eta_t}{2n(n-1)} \sum_{\substack{k,k' \in [n], \\ k \neq k'}} \|\nabla f(w_t; z_k, z_{k'}) - \nabla F_S(w_t)\|^2$$

$$= - \frac{\eta_t}{2} \|\nabla F_S(w_t)\|^2 + \frac{n(n-1)}{2b} \beta \eta_t^2 \left(1 + \frac{b-1}{n(n-1)}\right) \|\nabla F_S(w_t)\|^2$$

$$+ \frac{\eta_t}{2n(n-1)} \sum_{\substack{k,k' \in [n], \\ k \neq k'}} \|\nabla f(w_t; z_k, z_{k'}) - \nabla F_S(w_t)\|^2.$$

Then, taking expectation w.r.t. all randomness, we obtain that

$$\mathbb{E}[F_S(w_{t+1}) - F_S(w_t)]$$

$$\leq - \frac{\eta_t}{2} \mathbb{E}[\|\nabla F_S(w_t)\|^2] + \frac{n(n-1)}{2b} \beta \eta_t^2 \left(1 + \frac{b-1}{n(n-1)}\right) \mathbb{E}[\|\nabla F_S(w_t)\|^2]$$

$$+ \frac{\eta_t}{2n(n-1)} \sum_{\substack{k,k' \in [n], \\ k \neq k'}} \mathbb{E}[\|\nabla f(w_t; z_k, z_{k'}) - \nabla F_S(w_t)\|^2]$$

$$= \left(\frac{n(n-1)}{2b} \beta \eta_t^2 \left(1 + \frac{b-1}{n(n-1)}\right) - \frac{\eta_t}{2}\right) \mathbb{E}[\|\nabla F_S(w_t)\|^2]$$

$$+ \frac{\eta_t}{2n(n-1)} \sum_{\substack{k,k' \in [n], \\ k \neq k'}} \mathbb{E}[\|\nabla f(w_t; z_k, z_{k'}) - \nabla F_S(w_t)\|^2]$$

$$\leq - \frac{\eta_t}{4} \mathbb{E}[\|\nabla F_S(w_t)\|^2] + \frac{\eta_t}{2n(n-1)} \sum_{\substack{k,k' \in [n], \\ k \neq k'}} \mathbb{E}[\|\nabla f(w_t; z_k, z_{k'}) - \nabla F_S(w_t)\|^2]$$

$$\leq - \frac{\mu \eta_t}{2} \mathbb{E}[F_S(w_t) - F_S(w(S))] + \eta_t \Gamma(2\theta + 1) K^2,$$

where the second and the third inequalities are caused by $\eta_t \leq \frac{b}{2n(n-1)\left(1 + \frac{b-1}{n(n-1)}\right)\beta}$, Assumption 3.9 and Lemma C.3 (a). We can easily get

$$\mathbb{E}[F_S(w_{t+1}) - F_S(w(S))]$$

$$\leq \left(1 - \frac{\mu \eta_t}{2}\right) \mathbb{E}\left[F_S(w_t) - F_S(w(S))\right] + \Gamma(2\theta + 1) K^2 \eta_t$$

$$\leq \prod_{i=1}^{t} \left(1 - \frac{1}{2} \mu \eta_i\right) \mathbb{E}\left[F_S(w_1) - F_S(w(S))\right] + \sum_{i=1}^{t} \left(\Gamma(2\theta + 1) K^2 \eta_i\right)$$

$$\leq \prod_{i=1}^{t}\left(1-\frac{1}{2}\mu\eta_i\right)\mathbb{E}[F_S(w_1)-F_S(w(S))]+\eta_1\Gamma(2\theta+1)K^2\sum_{i=1}^{t}i^{-1}$$

$$\leq \prod_{i=1}^{t}\left(1-\frac{1}{2}\mu\eta_i\right)\mathbb{E}[F_S(w_1)-F_S(w(S))]+\eta_1\Gamma(2\theta+1)K^2\log(et),$$

where all inequalities are similar to the proof of Theorem 4.6. Similarly, we let $a_1 = 1 - \prod_{i=1}^{t}\left(1-\frac{1}{2}\mu\eta_i\right)$. With the help of Lemma C.1 (2), we further get

$$\mathbb{E}[\|\nabla f(w_t;z_{i_t},z_{j_t})\|^2]$$
$$\leq 2\beta\mathbb{E}[F_S(w_t)]$$
$$=2\beta\left((1-a_1)\mathbb{E}[F_S(w_1)]+a_1\mathbb{E}[F_S(w(S))]+\eta_1\Gamma(2\theta+1)K^2\log(et)\right).$$

To simplify the subsequent proof process, we let

$$\hat{\tau}(t)=2\beta\left((1-a_1)\mathbb{E}[F_S(w_1)]+a_1\mathbb{E}[F_S(w(S))]+\eta_1\Gamma(2\theta+1)K^2\log(et)\right).$$

Besides,

$$\|w_{t+1}-w'_{t+1}\|$$

$$=\left\|w_t-\frac{\eta_t}{b}\sum_{\substack{k,k'\in[n],\\k\neq k'}}\alpha_{t,k,k'}\nabla f(w_t;z_k,z_{k'})-w'_t+\frac{\eta_t}{b}\sum_{\substack{k,k'\in[n],\\k\neq k'}}\alpha_{t,k,k'}\nabla f(w'_t;z'_k,z'_{k'})\right\|$$

$$\leq\left\|w_t-w'_t-\frac{\eta_t}{b}\sum_{\substack{k,k'\in[n-2],\\k\neq k'}}\alpha_{t,k,k'}\nabla f(w_t;z_k,z_{k'})+\frac{\eta_t}{b}\sum_{\substack{k,k'\in[n-2],\\k\neq k'}}\alpha_{t,k,k'}\nabla f(w'_t;z_k,z_{k'})\right\|$$

$$+\frac{\eta_t}{b}\sum_{\substack{k\in\{n-1,n\},\\k'\in[n-2]}}\alpha_{t,k,k'}\|\nabla f(w_t;z_k,z_{k'})-\nabla f(w'_t;z'_k,z'_{k'})\|$$

$$+\frac{\eta_t}{b}\sum_{\substack{k\in[n-2],\\k'\in\{n-1,n\}}}\alpha_{t,k,k'}\|\nabla f(w_t;z_k,z_{k'})-\nabla f(w'_t;z'_k,z'_{k'})\|$$

$$+\frac{\eta_t}{b}\sum_{\substack{k,k'\in\{n-1,n\},\\k\neq k'}}\alpha_{t,k,k'}\|\nabla f(w_t;z_k,z_{k'})-\nabla f(w'_t;z'_k,z'_{k'})\|$$

$$\leq\|w_t-w'_t\|+\frac{\eta_t}{b}\sum_{\substack{k,k'\in[n-2],\\k\neq k'}}\alpha_{t,k,k'}\|\nabla f(w_t;z_k,z_{k'})-\nabla f(w'_t;z_k,z_{k'})\|$$

$$+\left(\frac{4\eta_t(n-2)}{b}\times 2+\frac{4\eta_t}{b}\right)\alpha_{t,n-1,n}\|\nabla f(w_t;z_{n-1},z_n)\|$$

$$\leq\|w_t-w'_t\|+\frac{\beta\eta_t}{b}\sum_{\substack{k,k'\in[n-2],\\k\neq k'}}\alpha_{t,k,k'}\|w_t-w'_t\|+\frac{4(2n-3)\eta_t}{b}\alpha_{t,n-1,n}\|\nabla f(w_t;z_{n-1},z_n)\|$$

$$=\left(1+\frac{\beta\eta_t}{b}\sum_{\substack{k,k'\in[n-2],\\k\neq k'}}\alpha_{t,k,k'}\right)\|w_t-w'_t\|+\frac{4(2n-3)\eta_t}{b}\alpha_{t,n-1,n}\|\nabla f(w_t;z_{n-1},z_n)\|,$$

Taking expectation w.r.t. $J_t$, we can get

$$\mathbb{E}_{J_t}[\|w_{t+1} - w'_{t+1}\|]$$
$$\leq \left(1 + \frac{(n-2)(n-3)\beta\eta_t}{b}\mathbb{E}_{J_t}\left[\alpha_{t,n-3,n-2}\right]\right)\|w_t - w'_t\|$$
$$+ \frac{4(2n-3)\eta_t}{b}\mathbb{E}_{J_t}\left[\alpha_{t,n-1,n}\right]\|\nabla f(w_t; z_{n-1}, z_n)\|$$
$$\leq (1 + \beta\eta_t)\|w_t - w'_t\| + \frac{8\eta_t}{n}\|\nabla f(w_t; z_{n-1}, z_n)\|.$$

We also take expectation w.r.t. all randomness to get

$$\mathbb{E}[\|w_{t+1} - w'_{t+1}\|] \leq (1 + \beta\eta_t)\mathbb{E}[\|w_t - w'_t\|] + \frac{8\eta_t}{n}\mathbb{E}[\|\nabla f(w_t; z_{n-1}, z_n)\|].$$

We further take summation from $t = 1$ to $T - 1$ and use Lemma C.2 (c) to obtain

$$\mathbb{E}[\|w_T - w'_T\|] \leq \sum_{t=1}^{T-1}\left(\prod_{k=t+1}^{T-1}(1 + \beta\eta_t)\right)\frac{8\eta_t}{n}\sqrt{\hat{\tau}(t)}$$
$$\leq \sum_{t=1}^{T-1}\left(\exp\left(\beta\sum_{k=t+1}^{T-1}\eta_t\right)\right)\frac{8\eta_t}{n}\sqrt{\hat{\tau}(T-1)}$$
$$\leq \exp\left(\beta\sum_{k=1}^{T-1}\eta_t\right)\frac{8}{n}\sqrt{\hat{\tau}(T-1)}\sum_{t=1}^{T-1}\eta_t$$
$$\leq \exp\left(\beta\eta_1\log(e(T-1))\right)\frac{8}{n}\sqrt{\hat{\tau}(T-1)}\eta_1\log(e(T-1))$$
$$\leq (e(T-1))^{\beta\eta_1}\frac{8}{n}\sqrt{\hat{\tau}(T-1)}\eta_1\log(e(T-1))$$
$$\leq \frac{8\eta_1}{n}(e(T-1))^{\beta\eta_1}\sqrt{\hat{\tau}(T-1)}\log(e(T-1)),$$

that is,

$$\mathbb{E}[\|w_T - w'_T\|]$$
$$\leq\mathcal{O}\left(\frac{1}{n}\beta^{-\frac{1}{2}}T^{\frac{b}{2n(n-1)\left(1+\frac{b-1}{n(n-1)}\right)}}\log T\sqrt{a_1\mathbb{E}[F_S(w(S))] + \beta^{-1}\Gamma(2\theta + 1)\log T}\right).$$

When $\mathbb{E}[F_S(w(S))] \leq \mathcal{O}(n^{-1})$,

$$\mathbb{E}[\|w_T - w'_T\|] \leq \mathcal{O}\left((\beta n)^{-1}T^{\frac{b}{2n(n-1)\left(1+\frac{b-1}{n(n-1)}\right)}}\Gamma(2\theta + 1)^{\frac{1}{2}}(\log T)^{\frac{3}{2}}\right).$$

□

**Proof of Corollary 4.10**: By Theorem 4.1 (b) and Theorem 4.9, we can get that

$$|\mathbb{E}[F(w_T) - F_S(w_T)]|$$
$$\leq 2\mathbb{E}[F_S(w_T)] + \frac{(4\theta)^\theta K}{n(n-1)}\sum_{\substack{i,j\in[n],\\i\neq j}}\mathbb{E}\left[\|w_T - w'_T\|\right]$$
$$\leq\mathcal{O}\left(\mathbb{E}[F_S(w_T)] + \frac{1}{n}\beta^{-\frac{1}{2}}T^{\frac{b}{2n(n-1)\left(1+\frac{b-1}{n(n-1)}\right)}}(4\theta)^\theta\log T\sqrt{a_1\mathbb{E}[F_S(w(S))] + \beta^{-1}\Gamma(2\theta + 1)\log T}\right).$$

When $\mathbb{E}[F_S(w(S))] \leq \mathbb{E}[F_S(w_T)] = \mathcal{O}\left(n^{-1}\right)$,

$$|\mathbb{E}[F(w_T) - F_S(w_T)]| = \mathcal{O}\left((\beta n)^{-1}T^{\frac{b}{2n(n-1)\left(1+\frac{b-1}{n(n-1)}\right)}}(4\theta)^\theta(\Gamma(2\theta + 1))^{\frac{1}{2}}(\log T)^{\frac{3}{2}}\right).$$

$\square$

**Proof of Theorem 4.11**: Without loss of generality, we can let $S_{i,j} = S_{n-1,n}$. and $J_t = \{(i_{t,1}, j_{t,1}), (i_{t,2}, j_{t,2}), ..., (i_{t,b}, j_{t,b})\}, t \in \mathbb{N}$. According to Lemma C.1 (1) and taking expectation w.r.t. $J_t$, we have

$$\mathbb{E}_{J_t}[F_S(w_{t+1}) - F_S(w_t)]$$

$$\leq \mathbb{E}_{J_t}\left[\langle w_{t+1} - w_t, \nabla F_S(w_t)\rangle + \frac{1}{2}\beta\|w_{t+1} - w_t\|^2\right]$$

$$= \mathbb{E}_{J_t}\left[-\frac{\eta_t}{b}\sum_{m=1}^{b}\langle\nabla f(w_t; z_{i_{t,m}}, z_{j_{t,m}}), \nabla F_S(w_t)\rangle + \frac{\beta\eta_t^2}{2}\left\|\frac{1}{b}\sum_{m=1}^{b}\nabla f(w_t; z_{i_{t,m}}, z_{j_{t,m}})\right\|^2\right]$$

$$\leq -\eta_t\|\nabla F_S(w_t)\|^2 + \beta\eta_t^2\mathbb{E}_{J_t}\left[\left\|\frac{1}{b}\sum_{m=1}^{b}\nabla f(w_t; z_{i_{t,m}}, z_{j_{t,m}}) - \nabla F_S(w_t)\right\|^2\right] + \beta\eta_t^2\|\nabla F_S(w_t)\|^2$$

$$\leq -\frac{1}{2}\eta_t\|\nabla F_S(w_t)\|^2 + \beta\eta_t^2\mathbb{E}_{J_t}\left[\|\nabla f(w_t; z_{i_{t,1}}, z_{j_{t,1}}) - \nabla F_S(w_t)\|^2\right]$$

$$\leq -\mu\eta_t\left(F_S(w_t) - F_S(w(S))\right) + \beta\eta_t^2\mathbb{E}_{J_t}\left[\|\nabla f(w_t; z_{i_{t,1}}, z_{j_{t,1}}) - \nabla F_S(w_t)\|^2\right],$$

where the third inequality is due to $\eta_t \leq \frac{b}{2n(n-1)\left(1+\frac{b-1}{n(n-1)}\right)\beta} \leq \frac{1}{2\beta}$ and the following inequality, i.e.,

$$\mathbb{E}_{J_t}\left[\left\|\frac{1}{b}\sum_{m=1}^{b}\nabla f(w_t; z_{i_{t,m}}, z_{j_{t,m}}) - \nabla F_S(w_t)\right\|^2\right]$$

$$\leq \frac{1}{b}\sum_{m=1}^{b}\mathbb{E}_{J_t}\left[\|\nabla f(w_t; z_{i_{t,m}}, z_{j_{t,m}}) - \nabla F_S(w_t)\|^2\right]$$

$$= \mathbb{E}_{J_t}\left[\|\nabla f(w_t; z_{i_{t,1}}, z_{j_{t,1}}) - \nabla F_S(w_t)\|^2\right].$$

Then, taking expectation w.r.t. all randomness, based on Lemma C.3 (a) and for any $m \in [b]$, we obtain that

$$\mathbb{E}[F_S(w_{t+1}) - F_S(w_t)]$$

$$\leq -\mu\eta_t\mathbb{E}[F_S(w_t) - F_S(w(S))] + \beta\eta_t^2\mathbb{E}\left[\|\nabla f(w_t; z_{i_{t,1}}, z_{j_{t,1}}) - \nabla F_S(w_t)\|^2\right]$$

$$\leq -\mu\eta_t\mathbb{E}[F_S(w_t) - F_S(w(S))] + 2\beta\eta_t^2\Gamma(2\theta + 1)K^2,$$

which is similar to the proof of Theorem 4.4. Therefore,

$$\mathbb{E}[F_S(w_{t+1}) - F_S(w(S))]$$

$$\leq (1 - \mu\eta_t)\mathbb{E}[F_S(w_t) - F_S(w(S))] + 2\beta\eta_t^2\Gamma(2\theta + 1)K^2$$

$$= \left(1 - \frac{\mu\eta_1}{t}\right)\mathbb{E}[F_S(w_t) - F_S(w(S))] + 2\beta\eta_1^2\Gamma(2\theta + 1)K^2t^{-2}.$$

We multiply both sides of the above inequality by $t(t - \frac{1}{2}\mu\eta_1)$ and get

$$t\left(t - \frac{1}{2}\mu\eta_1\right)\mathbb{E}[F_S(w_{t+1}) - F_S(w(S))]$$

$$\leq \left(t - \frac{1}{2}\mu\eta_1\right)(t - \mu\eta_1)\mathbb{E}[F_S(w_t) - F_S(w(S))] + 2\beta\eta_1^2\Gamma(2\theta + 1)K^2.$$

Then, we can take a summation from $t = 1$ to $T - 1$ to get

$$(T - 1)\left(T - 1 - \frac{1}{2}\mu\eta_1\right)\mathbb{E}[F_S(w_T) - F_S(w(S))]$$

$$\leq \left(1 - \frac{1}{2}\mu\eta_1\right)(1 - \mu\eta_1)\mathbb{E}[F_S(w_1) - F_S(w(S))] + 2\beta\eta_1^2\Gamma(2\theta + 1)K^2(T - 1)$$

$$\leq \left(1 - \frac{1}{2}\mu\eta_1\right)(1 - \mu\eta_1)\,\mathbb{E}[F_S(w_1) - F_S(w(S))] + 2\beta\eta_1^2\Gamma(2\theta + 1)K^2(T - 1).$$

Therefore,

$$\mathbb{E}[F_S(w_T) - F_S(w(S))]$$
$$\leq \frac{(1 - \frac{1}{2}\mu\eta_1)(1 - \mu\eta_1)}{(T - 1)(T - 1 - \frac{1}{2}\mu\eta_1)}\mathbb{E}[F_S(w_1)] + \frac{2\beta\eta_1^2\Gamma(2\theta + 1)K^2}{(T - 1 - \frac{1}{2}\mu\eta_1)}$$
$$= \mathcal{O}\left(\frac{1}{T^2} + \frac{\Gamma(2\theta + 1)}{\beta T}\right).$$

We finish the proof of optimization error bound.

Now, we prove excess risk bound. By Corollary 4.10, we get that

$$|\mathbb{E}[F(w_T) - F_S(w_T)]|$$
$$\leq \mathcal{O}\left(\mathbb{E}[F_S(w_T)] + \frac{1}{n}\beta^{-\frac{1}{2}}T^{\overline{2n(n-1)\left(1 + \frac{b-1}{n(n-1)}\right)}}(4\theta)^\theta \log T \sqrt{a_1\mathbb{E}[F_S(w(S))] + \beta^{-1}\Gamma(2\theta + 1)\log T}\right).$$

Then, considering $\mathbb{E}[F_S(w(S))] \leq \mathbb{E}[F_S(w_T)] = \mathcal{O}\left(n^{-1}\right)$, we have

$$|\mathbb{E}[F(w_T) - F(w^*)]|$$
$$\leq |\mathbb{E}[F_S(w_T) - F_S(w(S))]| + |\mathbb{E}[F(w_T) - F_S(w_T)]|$$
$$\leq \mathcal{O}\left(\frac{1}{T^2} + \frac{\Gamma(2\theta + 1)}{\beta T} + \mathbb{E}[F_S(w_T)]\right.$$
$$\left. + \frac{1}{n}\beta^{-\frac{1}{2}}T^{\overline{2n(n-1)\left(1 + \frac{b-1}{n(n-1)}\right)}}(4\theta)^\theta \log T \sqrt{a_1\mathbb{E}[F_S(w(S))] + \beta^{-1}\Gamma(2\theta + 1)\log T}\right)$$
$$\leq \mathcal{O}\left(\frac{\Gamma(2\theta + 1)}{\beta T} + (\beta n)^{-1}T^{\overline{2n(n-1)\left(1 + \frac{b-1}{n(n-1)}\right)}}(4\theta)^\theta(\Gamma(2\theta + 1))^{\frac{1}{2}}(\log T)^{\frac{3}{2}}\right).$$

The proof is completed. $\qquad\square$

## C.7 DISCUSSIONS ABOUT OUR RESULTS

Table 4: Comparisons of our results ($\sqrt{}$-the reference has such a property; $\times$-the reference hasn't such a property; Cor.-Corollary; Thm.-Theorem; $L$-the parameter of Lipschitz continuity; $\beta$-the parameter of smoothness; $\theta$-the tail parameter of heavy-tailed gradient noise; $b' = \frac{b}{2n(n-1)\left(1 + \frac{b-1}{n(n-1)}\right)}$). Note that smoothness, as an indispensable assumption for the whole paper, is not included in this table.

| Algorithm | Assumptions | | | Generalization | Optimization |
|---|---|---|---|---|---|
| | $L$ | $\mu$ | $\theta$ | | |
| SGD (Cor. 4.3) | $\sqrt{}$ | $\times$ | $\times$ | $\mathcal{O}\left((\beta n)^{-1}L^2 T^{\frac{1}{2}}\log T\right)$ | — |
| SGD (Cor. 4.5) | $\times$ | $\times$ | $\sqrt{}$ | $\mathcal{O}\left((\beta n)^{-1}(4\theta)^\theta(\Gamma(2\theta + 1))^{\frac{1}{2}}T^{\frac{1}{2}}(\log T)^{\frac{3}{2}}\right)$ | — |
| SGD (Cor. 4.7, Thm. 4.8) | $\times$ | $\sqrt{}$ | $\sqrt{}$ | $\mathcal{O}\left((\beta n)^{-1}(4\theta)^\theta(\Gamma(2\theta + 1))^{\frac{1}{2}}T^{\frac{1}{4}}(\log T)^{\frac{3}{2}}\right)$ | $\mathcal{O}\left(\frac{\Gamma(2\theta + 1)}{\beta T}\right)$ |
| Minibatch SGD (Cor. 4.10, Thm. 4.11) | $\times$ | $\sqrt{}$ | $\sqrt{}$ | $\mathcal{O}\left((\beta n)^{-1}(4\theta)^\theta(\Gamma(2\theta + 1))^{\frac{1}{2}}T^{b'}(\log T)^{\frac{3}{2}}\right)$ | $\mathcal{O}\left(\frac{\Gamma(2\theta + 1)}{\beta T}\right)$ |

Table 4 presents all our generalization and optimization results for four different cases. The first case considers the general non-convex pairwise SGD involving bounded gradient condition. Its generalization bound is comparable to or even better than some previous results for pairwise SGD

with non-convex loss functions (see Table 1). The second case introduces the heavy-tailed gradient noise condition to remove the Lipschitz continuity assumption. Due to the non-convexity of the loss function, there is no way to ensure the access to a global minimizer, which is the reason why we don't study the optimization error bounds of the first two cases. The third case further considers the gradient dominance condition, a common condition in non-convex optimization, to get the sharper bounds for non-convex, heavy-tailed pairwise SGD in terms of $\ell_1$ on-average model stability tool. In the fourth case, we extend the analysis of the third case to the minibatch SGD to derive the first stability-based near-optimal bounds.

### C.7.1 DISCUSSIONS ON SOME DEPENDENCIES OF OUR RESULTS

**Dependencies on $T$.** From Corollary 4.5 to 4.7, we can find that the dependence on $T$ is improved from $T^{1/2}$ to $T^{1/4}$. The key of this transition is the upper bound of step size $\eta_1$. For Corollary 4.5, we set $\eta_1 \leq \frac{1}{2\beta}$ with the reason that the term $\|\nabla F_S(w_t)\|^2$ can be removed without any cost, which can be found in the second inequality of Appendix C.4. For Corollary 4.7, the upper bound of $\eta_1$ is set to be tighter than Corollary 4.5. In this case, if we directly remove $\|\nabla F_S(w_t)\|^2$, the bound is equal to Corollary 4.5, which is meaningless. So we introduce the additional assumption, PL condition, to decompose $\|\nabla F_S(w_t)\|^2$. The bound of Corollary 4.7 demonstrates that the tighter upper bound of $\eta_1$ combined with PL condition can lead to the tighter stability bound.

**Dependencies on $\beta$.** For all results (from Theorem 4.2 to Theorem 4.11), the dependencies are $\beta^{-1}$, which are similar and even tighter than Shen et al. (2019); Lei et al. (2021b)

**Dependencies on $\theta$.** In our work, sub-Weibull gradient noise assumption is introduced to derive the monotonic dependence of the bound on $\theta$, which is consistent with the papers we mentioned (Nguyen et al., 2019; Hodgkinson & Mahoney, 2021). However, inspired by Raj et al. (2023b), our dependence on $\theta$ essentially belongs to the dependence of the variance of loss function on $\theta$. Except for the variance, there are also some dependences of other parameters on $\theta$, such as the smoothness parameter, which need to be developed. These dependences may be not-monotonic as Raj et al. (2023a;b). Besides, other heavy-tailed distributions (such as $\alpha$-stable distributions Raj et al. (2023a;b)) will be considered in our future work to further explore the relationship between heavy tails and generalization performance.

**Dependencies on $\mu$.** For the results from Theorem 4.6 to Theorem 4.11, there is a dependence $1 - \prod_{i=1}^t \left(1 - \frac{1}{2}\mu\eta_i\right)$ on the PL parameter $\mu$. Obviously, this dependence is less than 1 and has a decreasing trend as $\mu$ decreases. When $\mathbb{E}[F_S(w(S))] = n^{-1}$, this dependence can be omitted.

### C.7.2 DETAILED COMPARISONS WITH LEI ET AL. (2021B)

In the main text, we mainly compare our results with (Lei et al., 2021b). It is also necessary to make detailed comparisons from other aspects, such as tools, assumptions, algorithms.

**1)Different stability tools:** We uses $\ell_1$ on-average model stability instead of uniform stability of Lei et al. (2021b).

**2)Different assumption:** We make a sub-Weibull gradient noise assumption to remove Lipschitz condition, which is one of our main contributions. While Lei et al. (2021b) isn't consider it. Besides, we compare the bound with PL condition and the one without PL condition to theoretically analyze the effect of PL condition.

**3)Minibatch SGD:** Our analysis for SGD is extended to the minibatch case, while Lei et al. (2021b) isn't consider it.

**4)Better expectation bounds:** Lei et al. (2021b) provided a uniform stability bound $\mathcal{O}\left((\beta n)^{-1} L^2 T^{\frac{\beta c}{\beta c+1}}\right)$, where the constant $c = \frac{1}{\mu}$ ($\mu$ is the parameter of PL condition). In general, $\mu$ is typically a very small value (Examples 1 and 2 in Lei & Ying (2021)) which leads to a large value of $c$. Thus, $T^{\frac{\beta c}{\beta c+1}}$ is closer to $T$ than the dependencies of our bounds on $T$. In other words, our bounds are tighter than Lei et al. (2021b).

**5)High probability bound:** Our proof can developed to establish high probability bounds which are provided in Appendix C.8. The orders of these high probability bounds are similar to our previous bounds in expectation.

### C.7.3   COMPARISONS WITH THE RESULTS OF POINTWISE LEARNING

Except for the comparisons with the related bounds for pairwise SGD, it is necessary to make some comparisons with some current results of pointwise SGD. We find that our results can be directly extended to the case of pointwise learning. Therefore, we provide the comparisons with some stability bounds of non-convex pointwise learning (Hardt et al. (2016); Zhou et al. (2022) in **Vs. Theorem 4.2**) and make a discussion about the non-convex stability-based generalization work with heavy tails ((Raj et al., 2023a;b) in **Vs. Theorems 4.4, 4.6, 4.9**).

**Vs.   Theorem 4.2** Hardt et al. (2016) developed the uniform stability bound $\mathcal{O}\left((\beta n)^{-1}L^{\frac{2}{\beta c+1}}T^{\frac{\beta c}{\beta c+1}}\right)$ under similar conditions, where the order depends on the smoothness parameter $\beta$ and a constant $c$ related to step size $\eta_t$. If $\log T \leq L^{\frac{2}{\beta c+1}-1}T^{\frac{\beta c}{\beta c+1}-\frac{1}{2}}$, the bound of Theorem 4.2 is tighter than theirs. A stability bound $\mathcal{O}\left((nL)^{-1}\sqrt{L+\mathbb{E}_S[v_S^2]}\log T\right)$ (Zhou et al., 2022) was established in the pointwise setting. Although it is $T^{1/2}$-times larger than the bound of Theorem 4.2, (Zhou et al., 2022) made a more stringent limitation to the step size $\eta_t$, i.e., $\eta_t = \frac{c}{(t+2)\log(t+2)}$ with $0 < c < 1/L$. If we make the same setting, we will get a similar bound with (Zhou et al., 2022).

**Vs. Theorems 4.4, 4.6, 4.9** As far as we know, there is a gap for the non-convex stability-based generalization work under the sub-Weibull gradient noise setting in the pointwise learning. For other heavy-tailed distributions, e.g., $\alpha$-stable distributions, there are a few papers (Raj et al., 2023a;b) studied the stability-based generalization bounds and made the conclusion that the dependence of generalization bound on the heavy-tailed parameter is not monotonic. Especially, Raj et al. (2023b) analyzed the dependencies of several constants on heavy-tailed parameter. Inspired by Raj et al. (2023b), we will further study the dependencies of other parameters (e.g., smoothness parameter $\beta$) on $\theta$, except for the monotonic dependence of the variance of the gradient for the loss function on $\theta$ in our bounds.

### C.8   PROOF OF HIGH PROBABILITY BOUNDS

**Theorem C.5.** *Let $S, S'$ and $S_{i,j}$ be constructed as Definition 3.5. Assume that pairwise SGD $A$, associated with loss function whose gradient noise obeys $subW(\theta, K)$, is $\ell_1$ on-average model $\epsilon$-stable without expectation. Then, we have*

$$|F(A(S)) - F_S(A(S))| \leq (4\theta)^\theta K\epsilon + 2F_S(A(S)).$$

**Proof of Theorem C.7**: Similar with the proof of Theorem 4.1 (b), we can get

$$
\begin{aligned}
&|F(A(S)) - F_S(A(S))| \\
\leq & \frac{1}{n(n-1)} \sum_{\substack{i,j\in[n], \\ i\neq j}} |f(A(S_{i,j}); z_i, z_j) - f(A(S); z_i, z_j)| \\
\leq & \frac{1}{n(n-1)} \sum_{\substack{i,j\in[n], \\ i\neq j}} \left(|f(A(S_{i,j}); z_i, z_j) - F_{S_{i,j}}(A(S_{i,j})) - (f(A(S); z_i, z_j) - F_S(A(S)))| \right. \\
& + |F_{S_{i,j}}(A(S_{i,j})) - F_S(A(S))|\big) \\
\leq & \frac{1}{n(n-1)} \sum_{\substack{i,j\in[n], \\ i\neq j}} \left((4\theta)^\theta K\|A(S_{i,j}) - A(S)\|\right) + 2F_S(A(S))
\end{aligned}
$$

$$\leq (4\theta)^\theta K\epsilon + 2F_S(A(S)),$$

where $\epsilon$ denotes the on-average model stability bound without expectation. $\qquad\square$

**Theorem C.6.** *Given $S, S'$ and $S_{i,j}$ described in Definition 3.5, let $\{w_t\}$ and $\{w'_t\}$ be produced by (5) on $S$ and $S_{i,j}$ respectively, where $\eta_t = \eta_1 t^{-1}, \eta_1 \leq (2\beta)^{-1}$, and let the parameters $A(S) = w_T$ and $A(S_{i,j}) = w'_T$ after $T$ iterations. Assume that the loss function $f(w; z, z')$ is $\beta$-smooth. Under Assumption 3.8, for any $\delta \in (0, 1)$, the following inequality holds with probability $1 - \delta$*

$$\frac{1}{n(n-1)} \sum_{i,j\in[n],i\neq j} \mathbb{E}\left[\|w_T - w'_T\|\right]$$

$$\leq \mathcal{O}\left((\beta n)^{-1} T^{\frac{1}{2}} \log^\theta(1/\delta) \log T \left(g(\theta) + (g(2\theta))^{\frac{1}{2}} (\log T)^{\frac{1}{2}}\right)\right),$$

*where $g(\theta) = (4e)^\theta$ for $\theta \leq 1$ and $g(\theta) = 2(2e\theta)^\theta$ for $\theta \geq 1$.*

**Proof of Theorem C.8**: From Equation (8), the following inequality holds

$$\|w_{t+1} - w'_{t+1}\|$$

$$\leq (1 + \eta_t\beta)\|w_t - w'_t\| + \frac{8n-12}{n(n-1)}\eta_t \left(\|\nabla f(w_t; z_{i_t}, z_{j_t}) - \nabla F_S(w_t)\| + \|\nabla F_S(w_t)\|\right),$$

We will firstly consider the term $\|\nabla F_S(w_t)\|$. From Equality (9), we know that

$$F_S(w_{t+1}) \leq F_S(w_1) + \sum_{t'=1}^{t}\left(\frac{1}{2}\eta_{t'} + \beta\eta_{t'}^2\right)\|\nabla f\left(w_{t'}; z_{i_{t'}}, z_{j_{t'}}\right) - \nabla F_S\left(w_{t'}\right)\|^2,$$

where $\frac{1}{2}\eta_t(\nabla f\left(w_t; z_{i_t}, z_{j_t}\right) - \nabla F_S\left(w_t\right))^2 \sim subW\left(2\theta, \frac{1}{2}\eta_t K^2\right)$ and $\beta\eta_t^2(\nabla f\left(w_t; z_{i_t}, z_{j_t}\right) - \nabla F_S\left(w_t\right))^2 \sim subW(2\theta, \beta\eta_t^2 K^2)$. According to Lemma C.4, we get the following inequality with probability at least $1 - \delta$

$$\sum_{t'=1}^{t}\left(\frac{1}{2}\eta_{t'} + \beta\eta_{t'}^2\right)\|\nabla f\left(w_{t'}; z_{i_{t'}}, z_{j_{t'}}\right) - \nabla F_S\left(w_{t'}\right)\|^2$$

$$\leq K^2 g(2\theta) \log^{2\theta}(2/\delta) \sum_{t'=1}^{t}\left(\frac{1}{2}\eta_{t'} + \beta\eta_{t'}^2\right),$$

where $g(\theta) = (4e)^\theta$ for $\theta \leq 1$ and $g(\theta) = 2(2e\theta)^\theta$ for $\theta \geq 1$. Thus,

$$F_S(w_{t+1}) \leq F_S(w_1) + K^2 g(2\theta) \log^{2\theta}(2/\delta) \sum_{t'=1}^{t}\left(\frac{1}{2}\eta_{t'} + \beta\eta_{t'}^2\right)$$

$$\leq F_S(w_1) + K^2 g(2\theta) \log^{2\theta}(2/\delta)\left(\frac{1}{2}\eta_1 \log(et) + \beta\eta_1^2\right),$$

where the second inequality is due to Lemma C.2(b), (c). Similar with the second inequality of Lemma C.1, we have

$$\frac{1}{2\beta}\|\nabla F_S(w)\|^2 \leq F_S(w) - \inf_{w'} F_S(w') \leq F_S(w).$$

Thus,

$$\|\nabla F_S(w_t)\|$$

$$\leq \sqrt{2\beta F_S(w_t)} \leq \sqrt{2\beta\left(F_S(w_1) + K^2 g(2\theta) \log^{2\theta}(2/\delta)\left(\frac{1}{2}\eta_1 \log(e(t-1)) + \beta\eta_1^2\right)\right)}.$$

Let $\tau(\theta, t) = \sqrt{2\beta F_S(w_t)} \leq \sqrt{2\beta\left(F_S(w_1) + K^2 g(2\theta) \log^{2\theta}(2/\delta)\left(\frac{1}{2}\eta_1 \log(e(t-1)) + \beta\eta_1^2\right)\right)}$. Then,

$$\|w_T - w'_T\|$$

$$\leq (1 + \eta_{T-1}\beta)\|w_{T-1} - w'_{T-1}\| + \frac{8n-12}{n(n-1)}\eta_{T-1}\big(\|\nabla F_S(w_{T-1})\|$$

$$+ \|\nabla f(w_{T-1}; z_{i_{T-1}}, z_{j_{T-1}}) - \nabla F_S(w_{T-1})\|\big)$$

$$\leq (1 + \eta_{T-1}\beta)\|w_{T-1} - w'_{T-1}\| + \frac{8n-12}{n(n-1)}\eta_{T-1}\big(\tau(\theta, T-1)$$

$$+ \|\nabla f(w_{T-1}; z_{i_{T-1}}, z_{j_{T-1}}) - \nabla F_S(w_{T-1})\|\big)$$

$$\leq \frac{8n-12}{n(n-1)}(e(T-1))^{\beta\eta_1} \sum_{t=1}^{T-1} \eta_t \left(\|\nabla f(w_t; z_{i_t}, z_{j_t}) - \nabla F_S(w_t)\| + \tau(\theta, T-1)\right)$$

$$\leq \frac{8n-12}{n(n-1)}(e(T-1))^{\beta\eta_1} \left(\sum_{t=1}^{T-1} \eta_t \|\nabla f(w_t; z_{i_t}, z_{j_t}) - \nabla F_S(w_t)\| + \sum_{t=1}^{T-1} \eta_t \tau(\theta, T-1)\right)$$

$$\leq \frac{8n-12}{n(n-1)}(e(T-1))^{\beta\eta_1} \left(Kg(\theta)\log^{\theta}(2/\delta) \sum_{t=1}^{T-1} \eta_t + \tau(\theta, T-1) \sum_{t=1}^{T-1} \eta_t\right)$$

$$\leq \frac{8n-12}{n(n-1)}(e(T-1))^{\beta\eta_1}\eta_1 \left(Kg(\theta)\log^{\theta}(2/\delta)\log(e(T-1)) + \tau(\theta, T-1)\log(e(T-1))\right),$$

where the fifth inequality is derived by $\eta_t(\nabla f(w_t; z_{i_t}, z_{j_t}) - \nabla F_S(w_t)) \sim subW(\theta, \eta_t K)$ and Lemma C.4. Then, the $\ell_1$ on-average model stability without expectation is proofed completely. $\square$

**Corollary C.7.** *Under Assumptions 3.7 (b) and 3.8, for the pairwise SGD (5) with $T$ iterations, the following inequality holds with probability $1 - \delta, \delta \in (0, 1)$*

$$|F(w_T) - F_S(w_T)| \leq \mathcal{O}\left((\beta n)^{-1}(4\theta)^{\theta}T^{\frac{1}{2}}\log^{\theta}(1/\delta)\log T \left(g(\theta) + (g(2\theta))^{\frac{1}{2}}(\log T)^{\frac{1}{2}}\right) + F_S(w_T)\right).$$

**Proof of Corollary C.9**: This corollary can be directly derived by combining the above two theorems, so we omit its proof. $\square$

## D  OUTLINES OF ALGORITHMIC STABILITY

The concept of algorithmic stability analysis was put forward as early as the end of the 20th century (Rogers & Wagner, 1978), and has been used for understanding the generalization performance of learning algorithms (Bousquet & Elisseeff, 2002; Elisseeff et al., 2005; Rakhlin et al., 2005). Bousquet & Elisseeff (2002) proposed the hypothesis stability, error stability and uniform stability, and various variants are designed in the next 20 years (Lei & Ying, 2020; Shalev-Shwartz et al., 2010; Hardt et al., 2016; Liu et al., 2017; Kuzborskij & Lampert, 2018; Chen et al., 2018; Ramezani-Kebrya et al., 2018; Foster et al., 2019; Deng et al., 2021). Hardt et al. (2016) built the connection between the generalization error of a randomized algorithm and its stability, and prove that SGD is uniformly stable for both convex and non-convex optimization. Shalev-Shwartz et al. (2010) designed the on-average stability for non-trivial learning problems, and Kuzborskij & Lampert (2018) gave a similar definition of on-average stability, the first data-dependent notion of algorithmic stability, which allows us to study generalization performance with the joint consideration of the properties of the learning algorithm and data-generating distribution. Moreover, some novel definitions are introduced in to capture the stability of model parameter directly including uniform model stability (called uniform argument stability (Liu et al., 2017)) and on-average model stability (Lei & Ying, 2020). In addition, Deng et al. (2021) proposed local elastic stability as a new distribution-dependent stability to get exponential generalization bounds.

Except for the above common stabilities for pointwise learning, they can be extended to the case of pairwise learning. For example, Shen et al. (2019) and Yang et al. (2021) provided the definitions of uniform stability (10) and uniform model stability (11) for pairwise learning, respectively, whose definitions are listed as follows.

$$\sup_{z,\tilde{z}\in\mathcal{Z}} \mathbb{E}[|f(A(S); z, \tilde{z}) - f(A(S_i); z, \tilde{z})|] \leq \epsilon, \forall S, \bar{S} \in \mathcal{Z}^n, \forall i \in [n] \tag{10}$$

$$\mathbb{E}[\|A(S) - A(S_i)\|] \leq \epsilon, \forall S, \bar{S} \in \mathcal{Z}^n, \forall i \in [n] \tag{11}$$

