# OpenReview forum: "Learning Guarantees for Non-convex Pairwise SGD with Heavy Tails"
_ICLR.cc/2024/Conference — Submitted to ICLR 2024_

### Official Review · Reviewer_dGQ5 · 2023-10-30

**Soundness:** 3 good
**Presentation:** 3 good
**Contribution:** 2 fair
**Rating:** 5
**Confidence:** 3

**Summary:**

The authors study the algorithmic stability type results for non-convex, heavy-tailed pairwise SGD by investigating the generalization performance and optimization jointly. Many theoretical results are obtained under various assumptions, including the general non-convex, non-convex without Lipschitz condition, non-convex with PL condition settings for pairwise SGD, and non-convex minibatch pairwise SGD.

**Strengths:**

A sequence of theoretical results is achieved, and the proofs seem to be correct and the logic is reasonable. The authors also introduced some new definition for pairwise learning algorithm to be $\ell_{1}$ on-average model stable. The paper is well written.

**Weaknesses:**

First, the model setup is a bit misleading. In the title and abstract, the authors call the setting heavy tails and heavy-tailed. However, what the authors really study is the sub-Weibull tails (Definition 3.6) that excludes polynomial decays as contrast to many papers the authors cited in the paper. Although there is no unique definition of heavy-tailedness, many readers would assume that you are talking about polynomial decay noise while you actually did not. I suggest you at least mention in the abstract that you are working with sub-Weibull tails.

Second, the technical novelty and the necessity of working with sub-Weibull distributed gradient noise is not very convincing to me. The reason is that I understand that sub-Weibull type distribution can appear in the concentration type inequalities if you want to obtain some high probability guarantees. But this is not what the authors are doing in this paper. The authors simply use very standard $L^2$ type arguments to study the SGD. What I meant by that is that if you look carefully at the proofs in the appendix, the authors only need some assumption to bound the 2nd moment, instead of relying on the full definition of the sub-Weibull distribution as is described in Definition 3.6. That means all the existing proof techniques in SGD for finite variance setups can all be directly used in the paper. If all you need is an application of Lemma C.3. with $p=2$, i.e. a second-moment estimate, why don’t you simply assume that instead of your Definition 3.6. in your paper? Will all the results still go through? Essentially, Lemma C.3. with $p=2$ says that if the second-moment depends on the parameter $\theta$ in a certain way (where $\theta$ measures the heaviness of the tail), then it will also appear in the final results. In my view, the authors are not using the full information of sub-Weibull distribution, and there are numerous papers in the literature about SGD with finite variance, and hence to include heavy tails in the title and abstract and use that as a selling point is a bit misleading.

Third, maybe I didn’t read the paper carefully enough, but it is not clear to me whether the same setting has been studied for pointwise SGD. If the answer is yes, then the authors should highlight the technical novelty and difficulty to extend the results to pairwise setting, which to me does not seem to be very difficult. Moreover, the authors should compare the results with the pointwise setting. On the other hand, if the answer is no, then I am wondering why the authors do not study the pointwise setting, which is much more common and popular in the literature, and the authors should comment on whether similar results can hold for pointwise SGD.

**Questions:**

For the main results, it would be better if the authors can provide some discussions on the monotonic (or not) dependence of the bound on $\theta$, which measures the heaviness of the tail, and provide some insights.

On page 2, before you talk about developing previous analysis techniques to the heavy-tailed pairwise cases, you should also cite some works about algorithmic stability and generalization bounds for pointwise SGD with heavy tails from the literature.

On page 3, you wrote that it is demonstrated in Nguyen et al. (2019); Hodgkinson and Mahoney (2021) that the generalization ability of SGD may suffer from the heavy-tailed gradient noise. However, I recently came across two more recent papers Raj et al. (2023) “Algorithmic stability of heavy-tailed stochastic gradient descent on least squares” and Raj et al. (2023) “Algorithmic stability of heavy-tailed SGD with general loss functions” that seem to argue heavy tails of gradient noise can help with generalization. I suggest you add more citations and discussions.

Assuming the gradient of loss function being Lipschitz is very reasonable. But assuming the gradient and the loss function itself are both Lipschitz seems to be quite strong. It would be nice if you can add some examples and discussions about your Assumption 3.7.

In Assumption 3.8. and Assumption 3.9., it would be better for you to add a line or two to explain what the expectations are taken with respect to.

---

> ### Author Response · Authors · 2023-11-18
>
> We are grateful to you for your valuable comments and constructive suggestions. The modifications mentioned in our response are all displayed in red font in our new manuscript.
> Considering the strict upper limit of 9 pages for the main text of the submission, most modifications are shown in Appendix and we will demonstrate their specific locations.
>
> **Q1:** ... In the title and abstract, the authors call the setting heavy tails and heavy-tailed. However, what the authors really study is the sub-Weibull tails (Definition 3.6) ... I suggest you at least mention in the abstract that you are working with sub-Weibull tails.
>
> **A1:** Thanks for your constructive comments. In this work, we indeed assume the gradient noises in SGD have sub-Weibull tails. We have modified the related statements in the abstract of our new manuscript to avoid misunderstanding. Other heavy-tailed distributions (such as $\alpha$-stable distributions[1]) will be considered in our future work to further explore the relationship between heavy tails and generalization performance.
>
> [1]T. Nguyen, et al. First exit time analysis of stochastic gradient descent under heavy-tailed gradient noise. NeurIPS, 2019.
>
> ***
> **Q2:** ... I understand that sub-Weibull type distribution can appear in the concentration type inequalities if you want to obtain some high probability guarantees. But this is not what the authors are doing in this paper. ... In my view, the authors are not using the full information of sub-Weibull distribution. ...
>
> **A2:** Thanks. For our bounds in expectation, we use the two bounds of second-moment in Lemma C.3, which can be regarded as the dependence of the variance of loss function on the parameter $\theta$ of heavy tails. It should be noted that our current proof can be developed to establish high probability bounds with concentration inequalities, which has been demonstrated at the beginning of “MAIN RESULTS” section. Limited by the response time, we have only provided a relationship between generalization error and on-average model stability, which removes the expectation of Theorem 4.1 (b), and the corresponding high probability generalization bound for Theorem 4.4 in Appendix C.8 of our new manuscript. The orders of these high probability bounds are similar to our previous bounds in expectation. After the end of Rebuttal, we will provide the high probability bounds for all cases.
> ***

---

> ### Author Response · Authors · 2023-11-18
>
> ***
> **Q3:** ... it is not clear to me whether the same setting has been studied for pointwise SGD. If the answer is yes, then the authors should highlight the technical novelty and difficulty to extend the results to pairwise setting... if the answer is no, then I am wondering why the authors do not study the pointwise setting, ... and the authors should comment on whether similar results can hold for pointwise SGD.
>
> **A3:** As far as we know, there is a gap for the non-convex stability-based generalization work under the sub-Weibull gradient noise setting in pointwise learning. [2] utilized uniform convergence tools to study high probability guarantees of non-convex pointwise SGD under the sub-Weibull gradient noise setting. Therefore, we try to analyze non-convex pairwise SGD with algorithmic stability tools. Our analysis can be directly extended to the corresponding pointwise case. In the following, we make comparisons with some stability bounds of non-convex pointwise learning (such as [3,4]) as the following **Vs. Theorem 4.2** and a discussion about the non-convex stability-based generalization work with heavy tails [5,6] in **Vs. Theorems 4.4, 4.6, 4.9**.
>
> **Vs. Theorem 4.2** [3] developed the uniform stability bound $\mathcal{O}\left((\beta n)^{-1} L^{\frac{2}{\beta c + 1}} T^{\frac{\beta c}{\beta c + 1}}\right)$ under similar conditions, where the order depends on the smoothness parameter $\beta$ and a constant $c$ related to step size $\eta_t$. If $\log T \leq L^{\frac{2}{\beta c + 1} - 1} T^{\frac{\beta c}{\beta c + 1} - \frac{1}{2}}$, the bound of Theorem 4.2 is tighter than theirs. A stability bound $\mathcal{O}\left((nL)^{-1}\sqrt{L+\mathbb{E}_S[v_S^2]}\log T \right)$ [4] was established in the pointwise setting. Although it is $T^{1/2}$-times larger than the bound of Theorem 4.2, [4] made a more stringent limitation to the step size $\eta_t$, i.e., $\eta_t=\frac{c}{(t+2)\log(t+2)}$ with $0<c<1/L$. If we make the same setting, we will get a similar bound with [4].
>
> **Vs. Theorems 4.4, 4.6, 4.9** For the heavy-tailed distributions except for sub-Weibull distributions, e.g., $\alpha$-stable distributions, there are a few papers [5,6] studied the stability-based generalization bounds and made the conclusion that the dependence of generalization bound on the heavy-tailed parameter is not monotonic. Especially, [5] analyzed the dependencies of several constants on heavy-tailed parameter. Our monotonic dependence on heavy-tailed parameter $\theta$ essentially belongs to the dependence of the variance of the gradient for the loss function on heavy tails. Inspired by [5], we will further study the dependencies of other parameters (e.g., smoothness parameter $\beta$) on $\theta$, except for the monotonic dependence in our bounds.
>
> The above statements have been discussed in Appendix C.7.3 of our modified manuscript.
>
> [2]S. Li and Y. Liu. High probability guarantees for nonconvex stochastic gradient descent with heavy tails. ICML, 2022.
>
> [3]M. Hardt, et al. Train faster, generalize better: Stability of stochastic gradient descent. ICML, 2016.
>
> [4]Y. Zhou, et al. Understanding generalization error of SGD in non-convex optimization. Machine Learning, 2022.
>
> [5]A. Raj, et al. Algorithmic stability of heavy-tailed SGD with general loss functions. ICML, 2023b.
>
> [6]A. Raj, et al. Algorithmic stability of heavy-tailed stochastic gradient descent on least squares. ALT, 2023a.
>
> ***
> **Q4:** ... it would be better if the authors can provide some discussions on the monotonic (or not) dependence of the bound on $\theta$ ...
>
> **A4:** Thanks. In our work, the sub-Weibull gradient noise assumption is introduced to derive the monotonic dependence of the bound on $\theta$, which is consistent with the papers we mentioned [7]. However, inspired by [8], our dependence on $\theta$ essentially belongs to the dependence of the variance of loss function on $\theta$. Except for the variance, there are also some dependencies of other parameters on $\theta$, such as the smoothness parameter, which need to be developed. These dependencies may be not-monotonic as [8,9]. We have added some related discussions in Appendix C.7.1 of our modified manuscript.
>
> [7]T. Nguyen, et al. First exit time analysis of stochastic gradient descent under heavy-tailed gradient noise. NeurIPS, 2019.
>
> [8]A. Raj, et al. Algorithmic stability of heavy-tailed SGD with general loss functions. ICML, 2023b.
>
> [9]A. Raj, et al. Algorithmic stability of heavy-tailed stochastic gradient descent on least squares. ALT, 2023a.
> ***

---

> ### Author Response · Authors · 2023-11-18
>
> ***
> **Q5:** On page 2, before you talk about developing previous analysis techniques to the heavy-tailed pairwise cases, you should also cite some works about algorithmic stability and generalization bounds for pointwise SGD with heavy tails from the literature.
>
> **A5:** As mentioned in **A3**, there is a gap in the stability-based generalization work for pointwise SGD with sub-Weibull gradient noise. The most related works are [10,11,12]. [10] used uniform convergence tools to derive some high probability generalization bounds of non-convex pointwise SGD with sub-Weibull tails. [11] and [12] analyzed the stability-based generalization performance of pointwise SGD under $\alpha$-stable Levy process. The latter studied the case of non-convex loss function. We have cited these works before talking about developing previous analysis techniques to the heavy-tailed pairwise cases on page 2 in our modified main text.
>
> [10]S. Li and Y. Liu. High probability guarantees for nonconvex stochastic gradient descent with heavy tails. ICML, 2022.
>
> [11]A. Raj, et al. Algorithmic stability of heavy-tailed stochastic gradient descent on least squares. ALT, 2023a.
>
> [12]A. Raj, et al. Algorithmic stability of heavy-tailed SGD with general loss functions. ICML, 2023b.
>
> ***
> **Q6:** ... I recently came across two more recent papers ... that seem to argue heavy tails of gradient noise can help with generalization. ...
>
> **A6:** Thanks. These two papers [13,14] motivate us to further develop our work. In our work, the sub-Weibull gradient noise assumption is introduced to derive a monotonic relationship between the generalization error and heavy tails, which is consistent with the papers we mentioned. However, inspired by [13], our dependence on heavy tails essentially belongs to the dependence of the variance of the gradient for the loss function on heavy tails. Except for the variance, there are also some dependencies of other parameters, such as the smoothness parameter, on heavy tails. We have cited these two papers and added some related discussions in Appendix C.7.1 of our modified manuscript. In the future, we will further analyze these dependencies which may be non-monotonic.
>
> [13]A. Raj, et al. Algorithmic stability of heavy-tailed stochastic gradient descent on least squares. ALT, 2023a.
>
> [14]A. Raj, et al. Algorithmic stability of heavy-tailed SGD with general loss functions. ICML, 2023b.
>
> ***
> **Q7:** ... It would be nice if you can add some examples and discussions about your Assumption 3.7.
>
> **A7:** Some previous work assumed the gradient and the loss function itself are both Lipschitz [15,16]. The Lipschitz w.r.t. the loss function itself is quite strong. Therefore, we attempt to remove this assumption with the sub-Weibull gradient noise assumption, which is one of our main contributions. We have cited some examples and modified the related statements to emphasize this contribution behind Assumption 3.7 in our new manuscript.
>
> [15]M. Hardt, et al. Train faster, generalize better: Stability of stochastic gradient descent. ICML, 2016.
>
> [16]Y. Lei and Y. Ying. Fine-grained analysis of stability and generalization for stochastic gradient descent. ICML, 2020.
>
> ***
> **Q8:** In Assumption 3.8. and Assumption 3.9., it would be better for you to add a line or two to explain what the expectations are taken with respect to.
>
> **A8:** Thanks. For Assumption 3.8, $\mathbb{E}$ is the expectation w.r.t. the samples indexes $i_t, j_t$. For Assumption 3.9, the definition of PL condition should be $\|\nabla F_S(w)\|^2 \geq  2\mu \left(F_S(w)- F_S(w(S))\right)$ without expectation. We have made the related modifications in Assumptions 3.8 and 3.9.

---

> > ### Author Response · Authors · 2023-11-23
> >
> > Dear Reviewer dGQ5,
> >
> > We sincerely appreciate your insightful comments on our manuscript. Your valuable suggestions have been incorporated into our revision, and we are eager to receive your feedback.
> >
> > Best regards,
> >
> > Authors

---

### Official Review · Reviewer_dBNi · 2023-10-31

**Soundness:** 3 good
**Presentation:** 3 good
**Contribution:** 2 fair
**Rating:** 6
**Confidence:** 3

**Summary:**

The paper studies the generalization performance of pairwise SGD in the non-convex setting, and in the presence of heavy tailed gradient noise. The generalization error for any learning algorithm is first bounded in terms of the $\ell_1$ on-average stability under the bounded gradient assumption.  A similar relationship is derived for the SGD under the assumption of heavy tailed gradient noise (without bounded gradient assumption). Next, bounds on the  $\ell_1$ on- average stability are derived for pairwise SGD under the aforementioned assumptions, which lead to explicit bounds on the generalization error.  Furthermore, bounds on generalization error and excess risk are derived for pairwise SGD under the PL condition, and assuming heavy-tailed gradient noise. These bounds are also extended to pairwise minibatch SGD.

**Strengths:**

1. The paper is written well overall with a clear problem setup, notation and motivation. Moreover, the related work section is very thorough and puts into perspective the results of the paper.

2. In terms of novelty, I believe there are no existing stability based guarantees for pairwise SGD with heavy tailed gradient noise in the nonconvex setting. Moreover, the relationship between generalization error and $\ell_1$ on average stability (Theorem 4.1) seems new to my knowledge.

**Weaknesses:**

1. Currently, no proof outline (or sketch) is provided in the main text. This makes it difficult to understand the extent of the novelty in the ideas underlying the proof. From my understanding, the proof steps seem to build heavily upon existing ideas from the literature on nonconvex pairwise SGD based learning (Lei et al, 2021b).


2. The results are stated in expectation throughout, which is weaker than the "high probability" results that exist in the literature for similar learning problems that use stability based analysis.

**Questions:**

1. In Assumption 3.8, is the expectation over all sources of randomness, including $w_t$?

2. In Theorem 4.1 (b), it is not clear to me whether this applies to any learning algorithm A, or is specific to SGD? This is because of the gradient noise assumption made therein which suggests that it is for SGD.

---

> ### Author Response · Authors · 2023-11-18
>
> We are grateful to you for your valuable comments and constructive suggestions. The modifications mentioned in our response are all displayed in red font in our new manuscript.
> Considering the strict upper limit of 9 pages for the main text of the submission, most modifications are shown in Appendix and we will demonstrate their specific locations.
>
> **Q1:** Currently, no proof outline (or sketch) is provided in the main text. ... the proof steps seem to build heavily upon existing ideas from the literature on nonconvex pairwise SGD based learning (Lei et al, 2021b).
>
> **A1:** Thanks for your constructive comments. Firstly, for the convenience of understanding the extent of the novelty in the ideas underlying the proof, we have provided a proof sketch figure at the beginning of Appendix C.
>
> Next, the main differences between our results and the non-convex stability bound of [1] are listed as follows.
>
> **1)Different stability tools:** We use $\ell_1$ on-average model stability instead of uniform stability of [1].
>
> **2)Different assumption:** We make a sub-Weibull gradient noise assumption to remove Lipschitz condition, which is one of our main contributions. While [1] doesn’t consider it. Besides, we compare the bound with PL condition and the one without PL condition to theoretically analyze the effect of PL condition.
>
> **3)Minibatch SGD:** Our analysis for SGD is extended to the minibatch case, while [1] doesn’t consider it.
>
> **4)Better expectation bounds:** [1] provided a uniform stability bound $\mathcal{O}\left((\beta n)^{-1}L^2T^{\frac{\beta c}{\beta c+1}}\right)$, where the constant $c=\frac{1}{\mu}$ ($\mu$ is the parameter of PL condition). In general, $\mu$ is typically a very small value (Examples 1 and 2 in [2]) which leads to a large value of $c$. Thus, $T^{\frac{\beta c}{\beta c+1}}$ is closer to $T$ than the dependencies of our bounds on $T$. In other words, our bounds are tighter than [1].
>
> **5)High probability bounds:** Our proof can be developed to establish high probability bounds which have been added in Appendix C.8 of our modified manuscript. The orders of these high probability bounds are similar to our previous bounds in expectation.
>
> We have emphasized these differences in Appendix C.7.2 of our modified manuscript.
>
> [1]Y. Lei, et al. Generalization guarantee of SGD for pairwise learning. NeurIPS, 2021b.
>
> [2]Y. Lei and Y. Ying. Sharper generalization bounds for learning with gradient-dominated objective functions. ICLR, 2021.
>
> ***
> **Q2:** The results are stated in expectation throughout, which is weaker than the "high probability"...
>
> **A2:** Thanks for your constructive comments. Our current proofs are not only used to derive expectation bounds but also establish high probability bounds, which has been demonstrated at the beginning of “MAIN RESULTS” section. Limited by the response time, we have only provided a relationship between generalization error and on-average model stability, which removes the expectation of Theorem 4.1 (b), and the corresponding high probability generalization bound for Theorem 4.4 in Appendix C.8 of our new manuscript.  The orders of these high probability bounds are similar to our previous bounds in expectation. After the end of Rebuttal, we will give the high probability bounds for all cases.
>
> ***
> **Q3:** In Assumption 3.8, is the expectation over all sources of randomness, including $w_t$?
>
> **A3:** In Assumption 3.8, $\mathbb{E}$ is the expectation w.r.t. the samples indexes $i_t, j_t$. $w_t$ isn’t included in the expectation. We have made the related modification in Assumption 3.8.
>
> ***
> **Q4:** In Theorem 4.1 (b), it is not clear to me whether this applies to any learning algorithm A, or is specific to SGD?...
>
> **A4:** Thanks. Due to Assumption 3.8, Theorem 4.1 (b) is specific to SGD. We have modified the related statements in our main text.
> ***

---

> ### Author Response · Authors · 2023-11-23
>
> Dear Reviewer dBNi,
>
> Thank you for dedicating your time and effort to offer insightful comments. We have covered all your concerns in our responses. We are looking forward to your reply.
>
> regards,
>
> Authors

---

> > ### Comment · Reviewer_dBNi · 2023-11-23
> >
> > Sorry for the delayed response. I have read the authors response to my queries and I am satisfied with their response. I will keep my original score for this paper.

---

> > > ### Author Response · Authors · 2023-11-23
> > >
> > > Thanks for your recognition of our work and for helping us improve the manuscript!

---

### Official Review · Reviewer_6TFb · 2023-11-10

**Soundness:** 2 fair
**Presentation:** 2 fair
**Contribution:** 3 good
**Rating:** 5
**Confidence:** 3

**Summary:**

This paper studies generalization and excess risk bounds for pairwise learning with SGD under non-convex loss functions. The main tool for proving generalization is a notion of $\ell_1$ on-average stability, which is adapted to the pairwise setting. It is shown that the bounded gradient condition for the loss function can be relaxed when having sub-Weibull noise in SGD iterates, which captures potentially heavy-tailed noise. Further assuming a PL condition leads to an improved generalization bound along with an optimization guarantee which overall leads to an excess risk bound for pairwise learning with SGD.

**Strengths:**

Relaxing the Lipschitzness requirement for the loss function and covering heavy-tailed noise distributions can be a significant forward step for stability-based SGD generalization bounds. Furthermore, the improvement from $T^{1/2}$ to $T^{1/4}$ in the stability bounds is a major improvement.

**Weaknesses:**

* My main concern is with the readability and precision of the current submission. The main text seems to lack sufficient intuition on the techniques behind the improvements and how prior analyses are modified to achieve the refined rates. Some examples for improving readability:
    * What is the valid range of values for $c$ in Table 1? Is it allowed to simply let $c \to 0$? This is important in order to compare the $T$-dependence of the current stability bounds with the literature.
    * Have the stability bounds of this paper, i.e. $\tilde{O}(T^{1/2}/n)$ and $\tilde{O}(T^{1/4}/n)$ under an additional PL condition, been established in the pointwise setting under the same assumptions or are they completely new?
    * What is the intuition behind improving prior dependencies on $T$ to $T^{1/2}$ under smoothness and $T^{1/4}$ under PL and smoothness? Specifically, how does the PL allow a transition from $T^{1/2}$ to $T^{1/4}$?
    * What is the dependence of the bounds in Theorems and Corollaries 4.6 to 4.11 on $\mu$? Without any hidden dependencies, it seems that one can let $\mu \to 0$ to prove the same results without the PL condition. Similarly, it seems like the dependence on $\beta$ is hidden in most statements which might be useful to highlight.

* The bounds of this paper are in expectation, while many similar bounds in the literature are stated with high probability. It might be useful to add a discussion on the possibility of establishing high probability bounds, especially how such bounds would interact with the heavy-tailed noise of SGD.

**Questions:**

* It seems that a term $|\mathbb{E}[F_S(w(S))] - F(w^*)|$ is missing from the RHS of Equation (4).

* Why is SGD initialized at zero in Definition 3.1? Is this a fundamental limitation or is it possible to handle arbitrary initialization?

* Perhaps there could be a discussion on why Definition 3.5 is called $\ell_1$ on-average stability even though the $\ell_2$ norm is used in the definition.

---

> ### Author Response · Authors · 2023-11-18
>
> We are grateful to you for your valuable comments and constructive suggestions. The modifications mentioned in our response are all displayed in red font in our new manuscript.
> Considering the strict page limit, most modifications are shown in Appendix and we will demonstrate their specific locations.
>
> **Q1:** What is the valid range of values for $c$ in Table 1?...
>
> **A1:** Thanks for your constructive comments to benefit for the comparisons with current stability bounds in Table 1.
>
> Firstly, [1] provided a uniform stability bound $\mathcal{O}\left((\beta n)^{-1}L^{\frac{2}{\beta c+1}}T^{\frac{\beta c}{\beta c+1}}\right)$, where the constant $c$ is just set to be greater than 0. Thus, under the same assumption with [1], our bound (Theorem 4.2) is tighter than it when $c$ satisfies the following inequality
> $L^{\frac{2}{\beta c+1}} T^{\frac{\beta c}{\beta c+1}} \geq LT^{1/2}\log T$,
> i.e., $0<c\leq \frac{1}{\beta}\left(\frac{1}{\log_{L^2T^{-1}}(LT^{-1/2}\log T)}-1\right)$.
>
> Secondly, [2] provided a uniform stability bound $\mathcal{O}\left((\beta n)^{-1}L^2T^{\frac{\beta c}{\beta c+1}}\right)$, where the constant $c=\frac{1}{\mu}$ ($\mu$ is the parameter of PL condition). In general, $\mu$ is typically a very small value (Examples 1 and 2 in [3]) which leads to a large value of $c$. Thus, $T^{\frac{\beta c}{\beta c+1}}$ is closer to $T$ than $T^{1/2}\log T$ in our bound (Theorem 4.2). In other words, our bound is tighter than [2].
>
> The aforementioned detailed analysis has been added behind Theorem 4.2 of our modified manuscript to improve readability.
>
> [1]W. Shen, et al. Stability and optimization error of stochastic gradient descent for pairwise learning. arXiv, 2019.
>
> [2]Y. Lei, et al. Generalization guarantee of SGD for pairwise learning. NeurIPS, 2021b.
>
> [3]Y. Lei and Y. Ying. Sharper generalization bounds for learning with gradient-dominated objective functions. ICLR, 2021.
>
> ***
> **Q2:** Have the stability bounds of this paper been established in the pointwise setting ...?
>
> **A2:**
> As far as we know, there is a gap for the non-convex stability-based generalization work under the sub-Weibull gradient noise setting in pointwise learning. [4] utilized uniform convergence tools to study high probability guarantees of non-convex pointwise SGD under the sub-Weibull gradient noise setting. Therefore, we try to analyze non-convex pairwise SGD with algorithmic stability tools. Our results can be directly extended to the case of pointwise learning. In the following, we make comparisons with some stability bounds of non-convex pointwise learning [5,6] in **Vs. Theorem 4.2** and a discussion about the non-convex stability-based generalization work with heavy tails [7,8] in **Vs. Theorems 4.4, 4.6, 4.9**.
>
> **Vs. Theorem 4.2** [5] developed the uniform stability bound $\mathcal{O}\left((\beta n)^{-1} L^{\frac{2}{\beta c + 1}} T^{\frac{\beta c}{\beta c + 1}}\right)$ under similar conditions, where the order depends on the smoothness parameter $\beta$ and a constant $c$ related to step size $\eta_t$. If $\log T \leq L^{\frac{2}{\beta c + 1} - 1} T^{\frac{\beta c}{\beta c + 1} - \frac{1}{2}}$, the bound of Theorem 4.2 is tighter than theirs. A stability bound $\mathcal{O}\left((nL)^{-1}\sqrt{L+\mathbb{E}_S[v_S^2]}\log T \right)$ [6] was established in the pointwise setting. Although it is $T^{1/2}$-times larger than the bound of Theorem 4.2, [6] made a more stringent limitation to the step size $\eta_t$, i.e., $\eta_t=\frac{c}{(t+2)\log(t+2)}$ with $0<c<1/L$. If we make the same setting, we will get a similar bound with [6].
>
> **Vs. Theorems 4.4, 4.6, 4.9** For the heavy-tailed distributions except for sub-Weibull distributions, e.g., $\alpha$-stable distributions, there are a few papers [7,8] studied the stability-based generalization bounds and made the conclusion that the dependence of generalization bound on the heavy-tailed parameter is not monotonic. Especially, [7] analyzed the dependencies of several constants on heavy-tailed parameter. Our monotonic dependence on heavy-tailed parameter $\theta$ essentially belongs to the dependence of the variance of the gradient for the loss function on heavy tails. Inspired by [7], we will further study the dependencies of other parameters (e.g., smoothness parameter $\beta$) on $\theta$, except for the monotonic dependence in our bounds.
>
> We have added these statements in Appendix C.7.3 of our modified manuscript.
>
> [4]S. Li and Y. Liu. High probability guarantees for nonconvex stochastic gradient descent with heavy tails. ICML, 2022.
>
> [5]M. Hardt, et al. Train faster, generalize better: Stability of stochastic gradient descent. ICML, 2016.
>
> [6]Y. Zhou, et al. Understanding generalization error of SGD in non-convex optimization. Machine Learning, 2022.
>
> [7]A. Raj, et al. Algorithmic stability of heavy-tailed SGD with general loss functions. ICML, 2023b.
>
> [8]A. Raj, et al. Algorithmic stability of heavy-tailed stochastic gradient descent on least squares. ALT, 2023a.
> ***

---

> ### Author Response · Authors · 2023-11-18
>
> ***
> **Q3:** ... how does the PL allow a transition from $T^{1/2}$ to $T^{1/4}$?
>
> **A3:** Thanks. The key of the transition from $T^{1/2}$ to $T^{1/4}$ is the upper bound of step size $\eta_1$.
>
> For Theorem 4.4, we set $\eta_1 \leq \frac{1}{2\beta}$ with the reason that the term $\|\nabla F_S(w_t)\|^2$ can be removed without any cost, which can be found in the second inequality of Appendix C.4.
>
> For Theorem 4.6, the upper bound of $\eta_1$ is set to be tighter than Theorem 4.4. In this case, if we directly remove $\|\nabla F_S(w_t)\|^2$, the bound is equal to Theorem 4.4, which is meaningless. So we introduce the additional assumption, PL condition, to decompose $\|\nabla F_S(w_t)\|^2$. The bound of Theorem 4.6 demonstrates that the tighter upper bound of $\eta_1$ combined with PL condition can lead to the tighter stability bound. The above statement has been added to improve readability in Appendix C.7.1 of our modified manuscript.
>
> ***
> **Q4:** What is the dependence of the bounds in Theorems and Corollaries 4.6 to 4.11 on $\mu$? ... Similarly, it seems like the dependence on $\beta$ is hidden ...
>
> **A4:** **1)Dependence on $\mu$:**
> The dependence on $\mu$ is indeed hidden. After the modifications on Theorems and Corollaries 4.6 to 4.11, this dependence $1-\prod\limits_{i=1}^t \left(1-\frac{1}{2} \mu \eta_i\right)$ has been uncovered in our modified manuscript. With this dependence, we can not let $\mu \rightarrow 0$ to prove the same results without the PL condition due to the tighter upper bound of $\eta_1\leq \frac{1}{4\beta}$. **A3** provides the consideration about the upper bounds of step size $\eta_1$.
>
> **2)Dependence on $\beta$:**
> All dependencies on smoothness parameter $\beta$ are hidden in our results. In our modified manuscript, we have also uncovered them. For all results from Theorem 4.2 to Theorem 4.11, the dependencies are $\beta^{-1}$, which are similar to [9,10]. We have added the above discussions in Appendix C.7.1 of our new manuscript.
>
> [9]W. Shen, et al. Stability and optimization error of stochastic gradient descent for pairwise learning. arXiv, 2019.
>
> [10]Y. Lei, et al. Generalization guarantee of SGD for pairwise learning. NeurIPS, 2021b.
>
> ***
> **Q5:** ... It might be useful to add a discussion on the possibility of establishing high probability bounds ...
>
> **A5:** Thanks for your constructive comments. Our current proofs of the bounds in expectation can be developed to establish high probability bounds, which has been demonstrated at the beginning of “MAIN RESULTS” section. Limited by the response time, we have only provided a relationship between generalization error and on-average model stability, which removes the expectation of Theorem 4.1 (b), and the corresponding high probability generalization bound for Theorem 4.4 in Appendix C.8 of our new manuscript. The orders of these high probability bounds are similar to our previous bounds in expectation. After the end of Rebuttal, we will give the high probability bounds for all cases.
>
> ***
> **Q6:** It seems that a term $|\mathbb{E}[F_S(w(S))] - F(w^*)|$ is missing from the RHS of Equation (4).
>
> **A6:** The term $|\mathbb{E}[F_S(w(S))] - F(w^*)| = 0$ due to the expectation w.r.t. the training set $S$, which has been emphasized before Equation (4) to make it clearer.
>
> ***
> **Q7:** Why is SGD initialized at zero in Definition 3.1?
>
> **A7:** Since our proof can handle arbitrary initialization, we choose a simple initialization $w_1=0$ for convenience.
>
> ***
> **Q8:** ... why Definition 3.5 is called $\ell_1$ on-average stability...
>
> **A8:** Thanks. Our pairwise $\ell_1$ on-average model stability is defined following Definition 4 [11]. It should be noted that $\ell_1$ means the exponent of the term $\left\|A(S_{i,j}) - A(S)\right\|$ is 1, which is not related to the $\ell_2$ norm.
>
> [11]Y. Lei and Y. Ying. Fine-grained analysis of stability and generalization for stochastic gradient descent. ICML, 2020.

---

> > ### Comment · Reviewer_6TFb · 2023-11-23
> >
> > Thank you for your detailed response and elaboration. After reading the revision, I still agree with Reviewer dGQ5 in that I'm not sure why the contributions are stated for pairwise SGD even though they are novel for pointwise SGD, a setting that would attract much more attention from the community. I believe a new version of this submission with more highlights on technical novelties in comparison with prior work would further strengthen the work and help the readers, which is why I'm keeping my original score.
> >
> > On a side note, I'm not sure I completely understand why $\mathbb{E}[F_S(w(S))] = F(w^*)$ as $w(S)$ has a dependence on the training set $S$ and is not the population minimizer.

---

> ### Author Response · Authors · 2023-11-23
>
> Dear Reviewer 6TFb,
>
> As the author-reviewer discussion phase is nearing its conclusion, we would like to inquire if there are any remaining concerns or areas that may need further clarification. Your support during this final phase, especially if you find the revisions satisfactory, would be of great significance. We are looking forward to your reply.
>
> regards,
>
> Authors

---

> ### Author Response · Authors · 2023-11-23
>
> Dear Reviewer 6TFb,
>
> Thanks for checking our rebuttal and providing insightful inquiries!
>
> **For your first question “I'm not sure why the contributions are stated for pairwise SGD even though they are novel for pointwise SGD”:**
>
> >As we mentioned in **A3** of our response to Reviewer dGQ5, [1] had provided the generalization guarantees under the same setting of pointwise SGD with uniform convergence analysis. Although pointwise learning attracts much more attention than pairwise learning from the community, the latter has many applications, such as metric learning[2], ranking[3] and AUC maximication[4]. This is why we study the generalization guarantees of pairwise SGD. As for **“even though they are novel for pointwise SGD”**, our extension to pointwise SGD is not the first generalization analysis of pointwise SGD with sub-Weibull tails but the first stability-based generalization analysis. Except for the comparisons with some pairwise results, we made **some comparisons with some pointwise results including [1]** in our main text (in **Table 2, the Remarks of Corollary 4.5, Theorem 4.8, and Appendix C.7.3**).
>
> **For your second question “I'm not sure I completely understand why** $\mathbb{E}[F_S(w(S))] = F(w^*)$**:”**
>
> >We’re very sorry for the incorrect explanation. We would like to make a detailed decomposition, which follows the same idea as in [5], to correct it as follows:
> \begin{align}
> \left|\mathbb{E}_S[F(A(S)) - F(w^*)]\right|
> \leq & \left|\mathbb{E}_S[F(A(S)) - F_S(A(S)) + F_S(A(S)) - F_S(w(S)) + F_S(w(S)) - F_S(w^*) + F_S(w^*) - F(w^*)]\right|\\\\
> = & \left|\mathbb{E}_S[F(A(S)) - F_S(A(S)) + F_S(A(S)) - F_S(w(S)) + F_S(w(S)) - F_S(w^*)]\right|\\\\
> \leq & \left|\mathbb{E}_S[F(A(S)) - F_S(A(S)) + F_S(A(S)) - F_S(w(S))]\right|\\\\
> \leq & \left|\mathbb{E}_S[F(A(S)) - F_S(A(S))]\right| + \left|\mathbb{E}_S[F_S(A(S)) - F_S(w(S))]\right|,
> \end{align}
> where the first equality is due to $\mathbb{E}_S[F_S(w^*)] = F(w^*)$, and the second inequality is derived from $\mathbb{E}_S[F_S(w(S)) - F_S(w^*)] \leq 0$ and $\mathbb{E}_S[F(A(S)) - F(w^*)] \geq 0$.
>
> Please feel free to let us know if these address your concerns！
>
> regards,
>
> Authors
>
> [1]S. Li and Y. Liu. High probability guarantees for nonconvex stochastic gradient descent with heavy tails. ICML, 2022.
>
> [2]E. Xing, et al. Distance metric learning with application to clustering with side-information. NIPS, 2002.
>
> [3]W. Rejchel. On ranking and generalization bounds. Journal of Machine Learning Research, 2012.
>
> [4]Y. Ying, et al. Stochastic online AUC maximization. NIPS, 2016.
>
> [5]Y. Chen, et al. Stability and convergence trade-off of iterative optimization algorithms, 2018.

---

### Author Response · Authors · 2023-11-23
**Hope for more discussions**

Dear Reviewers,

Thanks for the comments of all reviewers to help us improve the manuscript. Considering the response deadline is approaching, we hope to have more discussions with you. If you have any further inquiries or suggestions, please do not hesitate to reach out to us.

regards,

Authors

---

### Meta-Review · Area_Chair_JnYF · 2023-12-14

**Metareview:**

In this paper, authors study pairwise learning with SGD under non-convexity and derive generalization/risk bounds. Authors rely on stability-based analysis and relax the bounded gradient assumption. They assume a PL condition and that the noise satisfies a sub-Weibull assumption in SGD, to be able to capture some sort of heavy-tailedness in the noise.

This paper was reviewed by 3 reviewers and received the following Rating/Confidence scores: 5/3, 6/3, 5/3. The reviewers had various concerns including:
1- readability and ambiguity in statements. 2- lack of intuition on the techniques 3-  model setup is not clearly introduced 3- Assumptions are not motivated well.

AC thinks that the paper has potential but requires significant revision, recommending reject for this ICLR. The decision is based upon the above weaknesses: None of the above weaknesses are severe when considered separately but together they decrease the quality of the paper significantly.

**Justification For Why Not Higher Score:**

There are various issues, e.g., 1- readability and ambiguity in statements. 2- lack of intuition on the techniques 3-  model setup is not clearly introduced 3- Assumptions are not motivated well.

**Justification For Why Not Lower Score:**

n/a

---

### Decision · Program_Chairs · 2024-01-16

Reject